# Physical measures of physical functioning as prognostic factors to predict outcomes in low back pain: A systematic review and narrative synthesis

Rameeza Rashed [ID]*, Afieh Niazigharemakher, David Walton, Katie Kowalski [ID], Alison Rushton [ID]

Health and Rehabilitation Sciences Graduate Program and School of Physical Therapy, London, Ontario, Canada

* rrashed2@uwo.ca

## Abstract

### Background

Low back pain (LBP) remains a major global health challenge. Effective management of LBP requires prognostic research to identify people at risk of poor outcome, enabling timely and targeted interventions.

### Objective

To synthesize the evidence for physical measures of physical functioning as prognostic factors for predicting outcome in LBP.

### Methods

This systematic review followed PRISMA and published protocol [PROSPERO-CRD42023406796] [1]. Searches were conducted in MEDLINE, EMBASE, CINAHL, Scopus and ProQuest Dissertations/Theses from inception to 29/5/2024. Hand searches of key journals and screening reference lists of included studies was performed. Prospective longitudinal studies, evaluating physical measures of physical functioning as prognostic factors, in adults 18years≥ with LBP and/or LBP-related leg pain were included. LBP related to malignancy, fracture, infection, cauda equina, inflammatory conditions, and measures; imaging, EMG, and motion capture with force plates or 3D video analysis were excluded. Two independent reviewers screened articles, extracted data, assessed risk of bias (RoB) using QUIPS. Due to high heterogeneity a narrative synthesis was conducted and GRADE determined the quality of evidence.

**Data availability statement:** All relevant data are within the paper and its Supporting Information files.

**Funding:** The author(s) received no specific funding for this work.

**Competing interests:** No authors have competing interests.

## Results

From 15,889 citations, 42 studies were included, with 50% assessed as high RoB. Low-quality evidence supports no predictive ability of high isometric back extension endurance, high handgrip strength, and high fingertip-to-floor test for good long term LBP outcomes. Very low-quality evidence supports inconsistent predictive ability of high lumbar extension range of motion and high straight leg raise range for good short-term outcomes, and high isometric back flexion endurance for good long-term LBP outcome. For studies that could not be synthesized, 41 physical measures of physical functioning were investigated, with 23 of them showing promising predictive ability for LBP outcome.

## Conclusion

This review highlights a lack of high-quality evidence regarding the predictive ability of physical measures of physical functioning in LBP. Findings indicate that the existing evidence is low-quality for no predictive ability and very low-quality for inconsistent predictive ability of physical measures of physical functioning. Low/very low-quality evidence suggests cautious interpretation. Imprecision, high RoB studies, and inadequately controlled confounding factors contributed to low/very low-quality evidence. This review also identifies emerging potential prognostic factors. An adequately powered, low RoB prospective longitudinal study using standardized measurement protocols and multivariable analysis is required to further investigate the promising predictive ability of physical measures of physical functioning in LBP. Future prognostic research should be grounded in strong theoretical rationale, including biological plausibility.

## Introduction

Low back pain (LBP) remains a major global health challenge, ranking among the top causes of years lived with disability [1,2]. Its impact extends beyond physical and mental health, imposing significant economic burdens [3–5]. Chronic LBP leads to ongoing medical expenses and indirect costs, such as lost work productivity [6]. Prognostic research identifies people at risk of poor outcome. Stratification based on prognostic factors facilitates personalized treatment plans, that would enhance effective LBP management [7]. A prognostic factor is any indicator that can predict subsequent health outcome and provides insights into the likely progression of a condition [8,9]. However, prognostic factors may not directly cause the outcome; rather, they can be markers or indicators of risk without being part of the causal pathway [10]. Causal factors always have some predictive value, but prognostic factors do not necessarily represent underlying causes. This study focuses exclusively on identifying and synthesizing prognostic factors that can predict LBP outcomes, rather than establishing causation.

Physical functioning is a fundamental aspect of health, defined by the Core Outcome Measures in Effectiveness Trials (COMET) Initiative as the impact of a disease

or condition on physical activities of daily living, such as walking and self-care [11,12]. It is recognized as a multidimensional construct encompassing several interconnected domains, including bodily structures and functions, performance of physical activities, as well as social and role-related participation [13]. Limitations in one domain may impact others, contributing to a decline in quality of life (QOL) [14].

Physical functioning can be assessed through different forms, including standardized self-report like the physical functioning subscale of the Short-Form 36 (SF-36) [15,16], can be directly observed by a rater (e.g., 6-minute walk test) [17], or can be quantified in real-world settings through wearable devices like accelerometers [18]. Each offer different insights into physical function, such as the patient's own self-perceptions in the case of self-rating scales, or activity in ecological settings in the case of accelerometers. The Initiative on Methods, Measurement, and Pain Assessment in Clinical Trials (IMMPACT) [18] recommends using both direct observation/quantification of activity in addition to participant self-report for a more fulsome evaluation of a participants' physical function [18]. In this systematic review, we are focused on physical measures of physical functioning that can predict outcomes in LBP.

Existing literature on prognostic factors in LBP has addressed a range of variables, e.g., psychological, personal and work-related factors [19,20]. However, there is a gap for the comprehensive investigation of physical measures of physical functioning. To date, two systematic reviews have investigated these factors. Hartvigsen et al. included physical measures evaluating physical functioning limited to low-tech clinical tests, and reported inconsistent evidence for various prognostic factors [21]. Verkerk et al. investigated a variety of prognostic factors, but did not comprehensively include physical measures of physical functioning. Their focus was solely on muscle endurance, strength, and aerobic capacity [22]. Both reviews also exhibited some methodological limitations, which contributed to low AMSTAR-2 criteria scores [23]. The AMSTAR-2 assessment for both reviews is provided in S1 File.

Despite extensive research on prognostic factors in LBP [24] there remains a significant gap in understanding the role of physical measures of physical functioning as prognostic factors for predicting LBP outcomes. Therefore, the purpose of this study was to comprehensively assess how physical measures of physical functioning can predict LBP outcomes.

## Objective

To synthesize the evidence for physical measures of physical functioning as prognostic factors predicting outcomes in the LBP population.

## Methods

### Design

This systematic review was designed using the PRISMA statement [25] and Cochrane Handbook [26]. It is registered in PROSPERO-(CRD42023406796) and follows a published protocol [27]. Our protocol initially included only English-language studies. However, with AI translation advancements, we translated non-English articles and validated the accuracy with bilingual individual familiar with the subject matter.

### Eligibility criteria (Informed by PICOS framework)

Inclusion and exclusion criteria informed by PICOS is summarized in Table 1.

Physical measures of physical functioning are categorized as the following:

1. Impairment-based measures: evaluating structure or function of a specific body part or system (e.g., range of motion) [15].

2. Performance-based measures: evaluating performance on a defined task in standardized environment (e.g., 6 min walk test) [18].

3. Activity in natural environment/real-world: evaluating activity in natural environment (e.g., accelerometery) [18].

**Table 1. Eligibility criteria (Informed by PICOS framework).**

| Inclusion criteria | Exclusion criteria |
|---|---|
| *Population (P)*. Participants aged 18 years and above with LBP and/or low back-related leg pain<br>*Potential physical prognostic factors (I)*. All physical measures of physical functioning that have been investigated as predictors of outcomes, and are practically feasible in terms of time, space, training, safety and cost to perform in hospital or community-based physiotherapy clinic were included. Physical measures of physical functioning are categorized as impairment-based measures, performance-based measures, activity in natural environment/real-world<br>*Comparator (C)*. Not applicable<br>*Outcome (O)*. Any outcome predicted by physical measures of physical functioning (intentional broad definition of outcome following scoping search that identified a limited number of studies) at any time point of follow up | Studies involving LBP related to malignancy, fracture, infection, cauda equina, rheumatoid arthritis, and ankylosing spondylitis were excluded.<br>Studies using measures such as imaging, electrophysiological measures (EMG), and motion capture gait analysis utilizing force plates or three-dimensional video analysis were excluded. |

## Information sources

A comprehensive search was performed from inception to May 29, 2024 on MEDLINE, EMBASE, CINAHL, and Scopus. Grey literature was searched using Open Grey System and ProQuest Dissertations and Theses. Hand searches of key journals (Spine, European Spine Journal, The Spine Journal) and screening reference list of included studies was also performed.

## Search strategy

The search strategy was developed in collaboration with a research librarian around the constructs of LBP, physical measures of physical functioning and prognostic factors. Search terms were informed by the "National Institute for Health and Care Excellence guidelines for LBP and sciatica in adults over the age of 16 year" [28], a previous systematic review [29], and a search filter developed to identify prognostic factor studies [30]. The search strategy developed in MEDLINE was adapted for use in other databases is provided in S2 File.

## Study selection process

The citations retrieved from searches were imported and archived into Covidence. This software detected and removed duplicate records. Based on eligibility criteria two authors [RR/AN] independently screened titles and abstracts, followed by full-texts screening. Discrepancies were resolved through discussion, and it was planned to consult a third reviewer (AR) if consensus was not achieved. Inter-rater reliability was assessed using Cohen's Kappa. [31,32].

## Data extraction process and items

Data were extracted by two independent authors using a standardized data extraction form, the Checklist for Critical Appraisal and Data Extraction for systematic reviews of prognostic factor studies (CHARMS-physical functioning) [33] which is an adapted version of the CHARMS checklist for primary studies of prediction models [9,34]. To ensure the reliability and feasibility of this modified form, pilot testing was performed. Data items extracted from each study were: LBP characteristics, participants, potential prognostic factors, outcome measure, results. For missing data, 5 authors were contacted via email as per mentioned in the protocol [27], response was received from 2 authors only [35,36].

## Risk of bias (RoB) in individual studies

To evaluate RoB, two independent authors used the QUIPS tool [37], recommended by the Cochrane Collaboration for assessing the RoB in prognostic studies [38]. The inter-rater reliability of QUIPS has been demonstrated to be acceptable, and previous studies have used QUIPS successfully in prognostic reviews [39,40]. It consists of multiple prompting items categorized into six domains: study participation, study attrition, prognostic factor measurement, outcome

measurement, study confounding, statistical analysis and reporting. Each domain is graded as low, moderate, or high risk of bias. Each study's overall RoB assessment was determined based on original QUIPS article and supporting studies [37,41,42]. Overall classification was low RoB if all domains were graded as low or one as moderate; high RoB if any domain was high or ≥3 were moderate; with all studies in between as moderate RoB [41]. Details of domains are provided in S3 File.

### Data synthesis and GRADE assessment

In line with the published protocol of this systematic review [27] and consistent with Cochrane handbook [26], narrative synthesis was planned a priori for circumstances of substantial heterogeneity. Due to high clinical, methodological and statistical heterogeneity [43], data were not pooled quantitatively. Clinically there was variability in LBP population characteristics, coexisting conditions, outcomes and follow up timepoints. Methodologically, almost half of the included studies were at high RoB, while others were at moderate or low RoB. Statistical heterogeneity was high as indicated by I² values >50% and reflected in wide variation in effect estimates across studies. This precluded a meaningful quantitative synthesis (meta-analysis), so a narrative synthesis was conducted [40]. According to the Cochrane Handbook, conducting a meta-analysis in the presence of substantial heterogeneity can produce misleading or clinically meaningless results and reduce the interpretability of the findings [44].

Narrative synthesis was based on prognostic factors, significant/nonsignificant association with different outcomes in relation to follow-up time points guided by Cochrane Consumers and Communication Review Group. Prognostic factors and outcome at short-term (<3 months), medium-term (≥3 months to <12 months), and long-term (≥12 months) were grouped and summarised when examined in ≥ 2 studies. The presence/absence and direction of an association between prognostic factors and outcomes at given time point was reported. If two studies reported the association in same direction then findings were considered consistent. Findings were considered inconsistent if studies reported associations in different directions or if they differed in statistical significance (achieved vs. not achieved), particularly when confidence intervals were not reported and only p-values were provided. Studies examining the same prognostic factor and outcome were narratively synthesized due to high clinical, methodological and statistical heterogeneity as meaningful meta-analysis was not possible. Bivariate analysis, odds ratios, beta coefficients, likelihood ratios, P values, confidence intervals (CI) chi-square test, narrative statements and multivariable analysis was reported per the original study.

Cumulative evidence was assessed by two authors independently, using modified GRADE proposed by Huguet et al. for prognostic factors research [45]. Cochrane recommends using GRADE to assess the quality of evidence in systematic reviews, including those with narrative syntheses when meta-analysis is not possible [46]. It is also recommended that narrative syntheses should provide structured summaries of findings using GRADE assessments to help interpret confidence in the evidence. The modified GRADE consists of six domains (phase of investigation, study limitations, inconsistency, indirectness, imprecision, publication bias) that determine certainty of evidence. In the modified GRADE for prognostic studies quality of evidence can be downgraded due to 5 factors [study limitations, inconsistency, indirectness, imprecision, and publication bias], and upgraded by 2 factors [moderate/large effect sizes (e.g., SMD 0.5–0.8, OR 2.5–4.25) and an exposure–response gradient]. Longitudinal designs are standard for prognostic research so study design is not significant feature in modified GRADE [47]. GRADE criteria used for determining the quality of evidence is provided in S4 File.

### Reporting bias

Reporting bias was evaluated by consistency to study protocols and published articles where available. Information of study protocols was obtained from included studies.

## Results

### Study selection

The search of 4 databases identified 15,889 citations and an additional 1,179 were identified from other sources. After removal of duplicates, 13,295 articles were screened by title and abstract, followed by full text screening of 488 articles. A total of 42 studies were included, with 2 articles by Nordeman et al. (2014, 2017) reported as 1 study. There were 9 non-English studies identified: 2 each (Japanese, German, Turkish language) and 1 each (French, Spanish and Chinese language). The list of non-English studies is provided in S5 File. Non-English studies were translated using an open-source software, Chat Generative Pre-Trained Transformer and validity of translation was cross checked by bilingual individual familiar with the subject matter. A PRISMA flow diagram [25] in **Fig 1** shows details of identified citations, selection and reasons for exclusion. At full text screening stage, reasons of exclusion are detailed in S6 File for each article. Inter-rater reliability between reviewers was 95.7% for title and abstract stage and 80.2% for full text screening stage, with Cohen's Kappa indicating fair agreement [48], however after discussion disagreement was resolved and complete agreement was achieved for each stage of screening.

### Study characteristics

The 42 included studies were published between 1989–2023, in 16 different countries with the greatest number of studies, coming from the United States (n = 8). The follow-up time range was 4 days to 30 months. The total number of participants

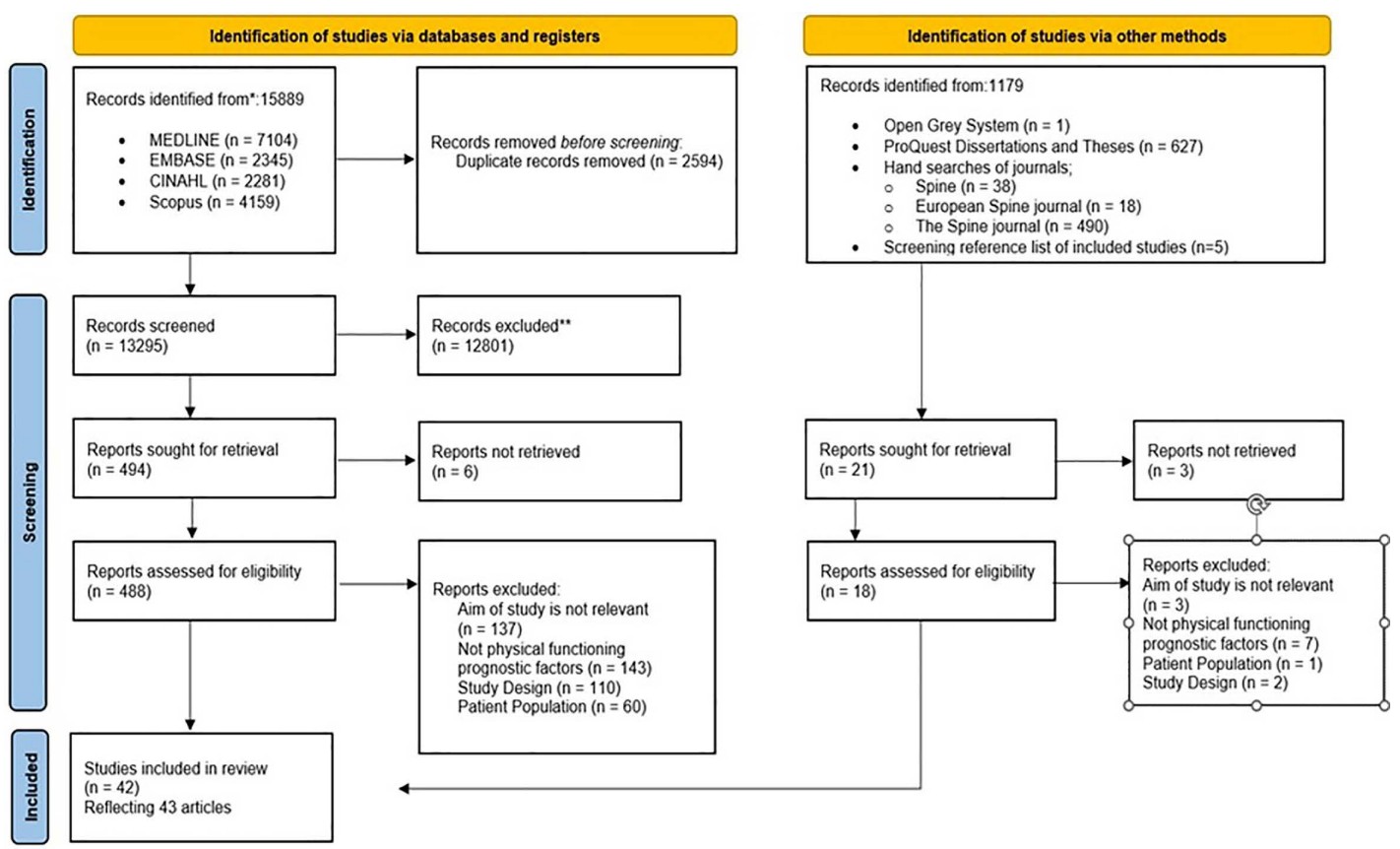

**Fig 1. PRISMA flow diagram.**

was 6,808 with sample sizes ranging from 24 to 675 participants. A total of 47 physical measures of physical functioning were assessed in the included studies. Details of included studies are provided in Table 2. Among the 42 studies, a total of 17 different outcomes were predicted. Disability was the most frequently evaluated in 15 studies, followed by pain in 13, return to work in 6, and non-recovery in 2 studies. Another 13 outcomes were assessed, each in only one study.

## RoB within studies

Of the 42 included studies, 21 (50%) studies were assessed as high RoB, 12 (29%) as low RoB, and 9 (21%) as moderate RoB provided in Table 3. The domain "study confounding" was the most rated as high RoB in 13 studies, due to not accounting for confounding factors in the analysis and the 'participation' domain was rated as low RoB in 33 studies, where the study samples fully represented the populations of interest. Details of each domain of QUIPS with reasons of high or low RoB is detailed in the S3 File.

## Results per physical prognostic factor of physical functioning

A total of 47 physical measures of physical functioning were investigated. Fingertip to floor test (FTF) was the most frequently evaluated measure in 7 studies followed by the back extension endurance test in 6 studies. Due to heterogeneity, only 6 measures could be synthesised across studies (≥2 studies) using GRADE (Table 4). Findings of 5 measures were based on bivariate analysis, while 1 measure was reported based on multivariable analysis. Overall quality of evidence using GRADE is shown in Table 5.

### Impairment-based measures

**Consistent findings.** *Fingertip to floor distance (FTF) with pain long-term:* Low-quality evidence (2 low RoB studies [49,50]) supports no statistically significant association between higher FTF (distance tip of middle finger to floor in flexion) and improved pain intensity (Visual Analogues Scale VAS, or Numeric pain rating Scale NRS) at 12 months. Therefore, low-quality evidence supports that higher FTF does not predict improved pain intensity long term.
*Handgrip strength with disability at long-term:* Low-quality evidence (2 low RoB studies [51,52] supports no statistically significant association between higher handgrip strength (maximum force exerted by hand muscles using hand-held dynamometer) and improved disability (Roland-Morris Disability Questionnaire – RMDQ) at 12 and 24 months. Therefore, low-quality evidence supports that higher handgrip strength does not predict improved disability long term.

**Inconsistent findings.** *Lumbar extension range of motion (ROM) with disability at short-term:* Very low-quality evidence (2 high RoB studies [35,53]) supports inconsistent statistically significant associations between higher lumbar extension ROM (distance between marks 10 cm above and 5 cm below the lumbosacral junction during active extension) and improved disability (Oswestry Disability Index-ODI and a self-reported disability questionnaire) at 2 and 4 weeks. Therefore, very low-quality evidence supports that higher lumbar extension ROM does not consistently predict improved disability short term.

*Straight Leg Raise-Range of Motion (SLR-ROM) with disability at short-term:* Very low-quality evidence (2 high RoB studies [54,55]) supports inconsistent statistically significant associations between SLR-ROM (degrees of hip flexion achieved through passive leg raising with a fully extended knee) and improved disability (Oswestry Disability Questionnaire-ODQ) at 2 and 4 weeks. Therefore, very low-quality evidence supports that higher SLR-ROM does not consistently predict improved disability short term.

### Performance-based measures

**Consistent findings.** *Isometric back extension endurance with pain at long term:* Low-quality evidence (1 low and 1 moderate RoB studies [49,56]) supports no statistically significant association between higher isometric extension

**Table 2.** Data extraction of 42 included studies.

| Characteristics of study | Objective of study | Characteristics of participants | Potential physical prognostic factors | Baseline measurement timepoint | Outcome and measure, Outcome assessment timepoints | Results |
|---|---|---|---|---|---|---|
| Berg et al. 2022 Netherland **Design:** Prospective longitudinal study **Funding source:** Not funded | To assess whether spinal stiffness, morning stiffness, ROM and LDD are prognostic factors for back pain after 1 year in older adults with back pain. | **N = 543** **Mean age (SD)** 67(8) years **Female** 320(59%) **Missing Data:** Not Reported **LBP characteristics** • Duration: < 1 week, 1 week to 6 weeks, 6 weeks to 3 months, and > 3 months • Distribution: Not Reported **Comorbidities:** Degenerative disease | **Impairment based measure** • Trunk side bending (reach further than knee or not) [Dichotomized (D)] • Trunk flexion < 10cm [D] **Equipment** Tape measure | Baseline | **Pain Intensity** (NRS) **Timepoints** 12 months (Long term) | **Bivariate Analysis-Univariate Logistic Regression Analysis** **Pain (NRS)** • Restricted lumbar side bending at baseline is significant as a prognostic factor for pain intensity at 12 months Crude: OR 2.1 P value 0.01) 95% CI (1.2–3.5) • Lumbar flexion < 10cm at baseline is not significant as prognostic factor for pain at 12 months Crude: OR 1.2 P value (0.41) 95% CI (0.8–1.8) **Multivariate logistic regression Analysis** • Restricted lumbar side flexion at baseline was not significant as a prognostic factor for pain intensity at 12 months Adjusted: OR 1.8 P value 0.07 95%CI (1.0; 3.2) • Lumbar anteflexion < 10cm at baseline was not significant as prognostic factor for pain at 12 months Adjusted: OR 1.4 P value 0.19 (95% CI 0.9–2.3) Notes: Adjusted for age, sex, BMI |
| Burton et al., 1991 UK **Design** Prospective longitudinal study **Funding source** Not Reported | To predict the 1- year clinical course of 109 patients with LBP trouble | **N = 109** **Mean age (SD) 41.8** (11.6) years **Female 50** (54%) **Missing Data:** Not Reported **LBP Characteristics:** Not Reported **Comorbidities:** Not Reported | **Impairment based measure** • Lumbar flexion ROM [Continuous (C)] • Lumbar extension ROM [C] **Performance based measures** • Sit up test [D] **Equipment** Flexicurve | Baseline | **Improved/ not improved** (pain and disability questionnaire) **Timepoints** 1m,3m,12 m (short term) | **Multivariate Analysis** **Improved/not improved pain and disability** • Higher lumbar flexion ROM at baseline was significant as prognostic factor for improved outcome (reduced pain and disability) at 1 month B = 0.47 • Lumbar extension ROM was not significant as prognostic factor for improved outcome (reduced pain and dyability) at 1 months (Narrative description is provided only) • Ability to do Sit up test at baseline is significant a prognostic factor for improved outcome (pain and disability) at 1 month B = 0.29 Notes: Individual variables were assessed at 1 month and then predictive batteries were tested at other timepoints) |

*(Continued)*

**Table 2.** (Continued)

| Characteristics of study | Objective of study | Characteristics of participants | Potential physical prognostic factors | Baseline measurement timepoint | Outcome and measure, Outcome assessment timepoints | Results |
|---|---|---|---|---|---|---|
| Campello et al., 2006 USA **Study Design** Prospective longitudinal study **Funding Source** Unfunded | To identify factors that predict work retention 24 months after treatment in patients with nonspecific LBP | N = 67 **Mean age (SD)** 40 (9.6) years **Female** 18 (2%) **Missing Data** n = 8 (8.7%) **LBP Characteristics** • Duration: Not Reported • Distribution: symptoms occurring primarily in the low back, with or without radiation to above knee level, not suggesting nerve root involvement **Comorbidities:** Not Reported | **Impairment based measures** • Gross Trunk flexibility [C] • Trunk flexion [C] • Trunk extension [C] • Treadmill walking test [C] **Performance-based measure** Progressive isointertial lifting evaluation [C] **Equipment** Treadmill Isokinetic device | Baseline | **Work retention** number of days that the subject worked during the 2-year follow-up period **Timepoints** 24 months (long term) | **Bivariate Analysis Work retention** • Higher gross Trunk flexibility at baseline is associated with higher retain to work at 24 months HR 2.47 95%CI 1.26–4.79 p 0.01 Trunk Flexion B 1.36 P value (0.01) 95%CI (1.24–4.38) Trunk extension B2.06 P value (0.04) 95%CI (1.02–4.16) • Isointertial lifting at baseline is not significant as prognostic factor for work retention at 24 months B 1.00 P value (0.65) 95%CI (0.99–1.02) • Treadmill walking at baseline is not associated with retain to work at 24 months. |
| Christensen et al., 1999 Denmark **Design** Prospective Longitudinal Study **Funding source** Grants from Sundhedspuljen | To identify prognostic factors associated with long-term prognosis | N = 330 **Mean age (SD)** 38(29) years **Female** 192 **Missing Data** n = 21 **LBP Characteristics** • Duration: acute pain 4 days (median) • Distribution: Pain in lower back only 60% Radiating to thigh 28% Pain below knee 12% | **Impairment based measures** • Restriction of lumbar movement (yes/No) [D] **Equipment** Not Reported | Baseline | **Sick leave** (Yes/No) **Functional recovery** (Yes/No) **Timepoints** 6 months 12 months (Medium, long term) | **Bivariate Analysis-Logistic Regression Analysis Sick leave and Functional recovery** • Presence of restriction of lumbar movement at baseline is significant as prognostic factor for poor outcome (on sick leave and not functionally recovered) at 6 months. OR 1.37 CI 95% (0.8–2.5) • Presence of restriction of lumbar movement at baseline is significant as prognostic factor for poor outcome (on sick leave and not functionally recovered) at 12 months. OR 1.40 CI 95% (0.8–2.6) **Multivariate Analysis – Logistic Regression Analysis** • Presence of restriction of lumbar movement at baseline is significant as prognostic factor for poor outcome (on sick leave and not functionally recovered) at 6 months. OR 1.30 CI 95% (0.7–2.4) • Presence of restriction of lumbar movement at baseline is significant as prognostic factor for poor outcome (on sick leave and not functionally recovered) at 12 months. OR 1.20 CI 95% (0.6–2.4) |

*(Continued)*

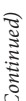

| Characteristics of study | Objective of study | Characteristics of participants | Potential physical prognostic factors | Baseline measurement timepoint | Outcome and measure, Outcome assessment timepoints | Results |
|---|---|---|---|---|---|---|
| Coste et al. 1994 France **Design**: Prospective longitudinal Study **Funding source:** Not Reported | To describe the natural course of recent acute low back pain in terms of morbidity (pain, disability) and absenteeism from work, and to evaluate the prognostic factors for these outcomes | **N = 103** **Mean (SD) age:** 46.5 (14.5) years **Female:** 41 (40%) **Missing data:** 11 (10%) **LBP characteristics:** • Duration: 7 days (median) with pain lasting ≤72 hours • Distribution: LBP without radiation below gluteal fold **Comorbidities:** Not Reported | **Impairment based measure** • Straight leg raise < 75° (SLR) [D] **Equipment** Goniometer | Baseline (First clinic visit for LBP complaint) | **Recovery:** Disappearance of pain (VAS) and disability (RMDQ) **Return to work:** Attendance at work **Timepoints** 3 Months (Short term) | **Bivariate Analysis; Survival Analysis-Log Rank Test** **Recovery** • SLR<75 degrees at baseline for LBP is not significant as prognostic factor for good outcome; recovery at 3 months P value (0.30) **Return to work** • SLR<75 degrees at baseline for LBP is not significant as prognostic factor for good outcome; return to work at 3 months P value (0.19) Note: SLR was not included in final prognostic model Cox proportional for Hazard Ratios. |
| Ekedahl et al. 2012 Sweden **Design**: Prospective longitudinal Study **Funding Source** Unfunded | To assess the predictive value of factors related to the change in RMDQ over 12 months using multivariate regression analysis. | **N = 65** **Mean (SD) age:** 45 (11) years **Female:**35(54%) **Missing data:**0% **LBP characteristics** • Duration: <6 weeks,6–13 weeks • Distribution: LBP without radiating pain below buttocks (n=38) LBP with radiating pain above knee (n=12) LBP with radiating pain below knee but without numbness (n=8) LBP with radiating pain, numbness and/or weakness (n=7) **Comorbidities:** Not Reported | **Impairment based measure** • SLR (angle between tibial crest and horizontal plane) [C] Finger to floor test (FTF)cm- [FTF BL & Change in FTF] [C] **Equipment** Goniometer | Baseline | **Disability** (RMDQ) Change in RMDQ **Timepoints** 1month 12 months (Short Long term) | **Multivariate Analysis- Linear Regression Analysis** **Change in RMDQ at 1 month** Entire sample (N=65) • Better FTF (FTF BL) at baseline is significant as prognostic factor for good outcome; reduced disability [change in RMDQ] over 1 month B 0.14 (0.06–0.21) P value (0.002) R² (0.23) • 1 month change in FTF (change in FTF) is significant as prognostic factor for good outcome; reduced disability [change in RMDQ] over 1 month B 0.26 (0.18–0.34) P value (0.001) R² (0.39) **Change in RMDQ at 12 months as outcome:** • Better FTF (FTF BL) at baseline is significant as prognostic factor for good outcome; reduced disability [change in RMDQ] over 12 months B 0.17 P value (<0.001) CI (0.09–0.26) R² (0.20) • 1 month change in FTF (change in FTF) is significant as prognostic factor for good outcome; reduced disability [change in RMDQ] over 12 months B 0.25 P value (< 0.001) CI (0.15–0.36) R² (0.27) |

*(Continued)*

| Characteristics of study | Objective of study | Characteristics of participants | Potential physical prognostic factors | Baseline measurement timepoint | Outcome and measure, Outcome assessment timepoints | Results |
|---|---|---|---|---|---|---|
| | | | | | | **Change in RMDQ at 1 month**<br>Radicular group (N=38)<br>• Better FTF at baseline (FTF) is significant as prognostic factor for good outcome; reduced disability [change in RMDQ] over 1 month. (short term)<br>B 0.13 P value (0.035) CI (0.01–0.25) $R^2$ [0.12]<br>• 1 month change in FTF (FTF change) is significant as prognostic factor for good outcome; reduced disability [change in RMDQ] over 1 month (short)<br>B 0.29 P value (<0.001) CI (0.18–0.40) $R^2$ [0.43]<br>**Change in RMDQ at 12 months**<br>• FTF at baseline (FTF BL) is significant as prognostic factor for good outcome; reduced disability [change in RMDQ] over 12 months.<br>B 0.15 P value (0.007) CI (0.04–0.26) $R^2$ (0.19)<br>• 1 month change in FTF is significant as prognostic factor for good outcome; reduced disability [change in RMDQ] over 12 months<br>B 0.23 P value (<0.001) CI (0.11–0.34) $R^2$ (0.31)<br>**Bivariate Analysis**<br>• SLR at baseline is not significant as prognostic factor for good outcome; reduced disability [change in RMDQ] over 1 month<br>P value (0.16)<br>• SLR at baseline is not significant as prognostic factor for good outcome; reduced disability [change in RMDQ] over 12 months<br>P value (>0.07)<br>NOTE: SLR is not included in Multivariate Analysis due to insignificant P value<br>**Radicular Group**<br>• SLR-ROM at baseline is not significant as prognostic factor for good outcome; reduced disability [change in RMDQ] over 1 month<br>P value (>0.18)<br>• SLR-ROM at baseline is not significant as prognostic factor for good outcome; reduced disability [change in RMDQ] over 12 months<br>P value (>0.06) |

*(Continued)*

| Characteristics of study | Objective of study | Characteristics of participants | Potential physical prognostic factors | Baseline measurement timepoint | Outcome and measure, Outcome assessment timepoints | Results |
|---|---|---|---|---|---|---|
| Enthoven et al., 2003 Sweden **Design:** Prospective longitudinal study **Funding Source** Not Reported | To examine associations between change over time in physical measures and changes in pain and disability | **N = 55** **Mean (SD) age:** 42 (14) Years **Female:** 29 (66%) **Missing Data n = 11** dropped out (20%) **LBP Characteristics** • Duration: Range <1 week to >3 months • Distribution: Full Back proved by movements **Comorbidities:** Not Reported | **Impairment based measures** • Thoracolumbar rotation-ROM [C] • Fingertip-to-floor distance (cm) [C] **Performance based measures** • Isometric endurance back flexors (sec) [C] Isometric endurance back extensors (sec) Modified Biering-Sørensen) [C] **Equipment** Inclinometer, Tape measure | Baseline 4 weeks | **Pain Intensity** VAS **Disability** ODI **Timepoints** 12 months (Long term) | **Bivariate Analysis- Simple Linear Regression** **Pain Intensity (VAS)** • Thoracolumbar rotation-ROM at baseline is not significant as prognostic factor for good outcome; reduced pain [VAS] at 12 months B −0.05 P value (>0.05) $R^2$ (0.00) • Higher Thoracolumbar rotation-ROM at 4 weeks is significant as prognostic factor for good outcome; reduced pain [VAS] at 12 months B −0.38 P value (<0.01) $R^2$ (0.16) • Fingertip-to-floor distance at baseline is not significant as prognostic factor for good outcome; reduced pain [VAS] at 12 months B −0.03 P value (>0.05) $R^2$ [0.00] • Lower Fingertip-to-floor distance at 4 weeks is significant as prognostic factor for good outcome; reduced pain [VAS] at 12 months B 0.98 P value (<0.01) $R^2$ (0.30) • Isometric endurance back flexors at baseline is significant as prognostic factor for good outcome; reduced pain [VAS] at 12 months B 0.13 P value<0.05 $R^2$ (0.09) • Isometric endurance back flexors at 4 weeks is not significant as prognostic factor for good outcome; reduced pain [VAS] at 12 months B −0.01 P value>0.05 $R^2$ (0.00) • Isometric endurance back extensors at baseline is not significant as prognostic factors for good outcomes; reduced pain [VAS] at 12 months B −0.00 P value (> 0.05) $R^2$ (0.00) • Higher Isometric endurance back extensors at 4 weeks is significant as prognostic factor for good outcome; reduced pain [VAS] at 12 months B-0.15 P value (<0.05) $R^2$ (−0.44) **Disability (ODI)** • Thoracolumbar rotation at baseline is not significant as prognostic factor for good outcome; reduced disability [ODI] at 12 months B 0.02 P value (>0.05) $R^2$ (0.00) • Higher Thoracolumbar rotation at 4 weeks is significant as prognostic factor for good outcome; reduced disability [ODI] at 12 months B −0.03 P value (<0.05) $R^2$ (0.21) • Fingertip-to-floor distance at baseline is not significant as prognostic factor for good outcome; reduced pain [VAS] at 12 months B −0.02 P value (>0.05) $R^2$ [0.00] |

| Characteristics of study | Objective of study | Characteristics of participants | Potential physical prognostic factors | Baseline measurement timepoint | Outcome and measure, Outcome assessment timepoints | Results |
|---|---|---|---|---|---|---|
| | | | | | | • Fingertip-to-floor distance at 4 weeks is significant as prognostic factor for good outcome; reduced pain [VAS] at 12 months B 0.80 P value (<0.01) R² [0.32] <br>• Isometric endurance back flexors at baseline is not significant as prognostic factor for good outcome; reduced disability [ODI] at 12 months B 0.07 P value (> 0.05) R² (0.04) <br>• Isometric endurance back flexors at 4 weeks is not significant as prognostic factor for good outcome; reduced disability [ODI] at 12 months B −0.02 P value (> 0.05) R² (0.00) <br>• Isometric endurance back extensors at baseline is not significant as prognostic factor for good outcome; reduced disability [ODI] at 12 months B −0.06 P value (> 0.05) R² (0.03) <br>• Higher Isometric endurance back extensors at 4 weeks is significant as prognostic factor for good outcome; reduced disability [ODI] at 12 months B (−0.14) P value (< 0.01) R² (0.17) Notes: Multivariate Analysis was not performed due to study limitations and limited number of participants |
| Felicio et al. 2017 Brazil **Design** Prospective longitudinal **Funding Source** Brazilian funding agencies Fundação de Amparo à Pesquisa do Estado de Minas Gerais (FAPEMIG) and Coordenação de Aperfeiçoamento de Pessoal de Nivel Superio [grant number 471264/2010–5]. | To examine whether HGS predicts disability in older women with acute LBP | **N = 135** **Mean (SD) age** 70(5) ± 5.4 years **Female:** 135 (100%) **Missing Data** n = 0 **LBP Characteristics** • Duration: Less than 6 weeks <br>• Distribution: LBP with or without leg radiating pain **Comorbidities** Not Reported | **Impairment based measure** • Hand Grip strength (HGS) [C] **Equipment** Jamar hydraulic hand dynamometer | Baseline | **Functional performance Disability** RMDQ] Brazilian Portuguese version of the 24-item **Physical capacity** Gait speed test **Timepoints** 12 months (Long term) | **Bivariate Analysis -Pearson correlation Disability (RMDQ)** • HGS at baseline is not significant as prognostic factor for good outcome; reduced disability at 12 months r −0.16 P value (0.053) **Gait speed** • Higher HGS at baseline is significant as prognostic factor for good outcome (lower time to walk a distance-higher gait speed) at 12 months r −0.24 P value (0.004) **Multivariable Linear Regression Analysis- Hierarchical linear regression model Gait speed** • Higher HGS at baseline is significant as prognostic factor for good outcome (lower time to walk a distance faster) at 12 months r −0.17 P value (0.043) |

*(Continued)*

**Table 2.** (Continued)

| Characteristics of study | Objective of study | Characteristics of participants | Potential physical prognostic factors | Baseline measurement timepoint | Outcome and measure, Outcome assessment timepoints | Results |
|---|---|---|---|---|---|---|
| Flynn et al. 2002 USA **Design** Prospective longitudinal study **Funding source**: Grant from foundation of physical therapy US | To develop a clinical prediction rule for identifying patients with low back pain who improve with spinal manipulation | **N = 75** **Mean age (SD)** 37.6(10.6) Years **Female** n = 29(41%) **Missing data** = 4 **LBP characteristics** • Duration: Not Reported • Distribution: Lumbosacral region Chief complaint of pain and/or numbness in the lumbar spine, buttock, and/or lower extremity **Comorbidities**: Not Reported | **Impairment-based measure** • Combined Hip Internal rotation range of motion > 35 (degree) [C] **Equipment** Not Reported | Baseline (first session) | **Disability** OSW [Success is improvement > 50%] **Timepoints** Second session (2–4) days (short) | **Multivariate Analysis-Logistic regression analysis** **Disability** • Higher Hip internal rotation range of motion > 35degree at baseline is significant as prognostic factor at first session for good outcome [low OSW] at second session LL3.25 95%CI (1.44,7.33) |
| Ghent et al. 2020 Australia **Design** Prospective longitudinal study **Funding** Not Reported | To expand outcome assessment in LDH surgery to include gait metrics from wearable devices. | **N = 24** **Mean (SD) age**: 49.6 (15.1) years. **Female n (%)**:10 (41%) **Missing data:** 0 **LBP characteristics:** Not Reported • Duration: Not Reported **Comorbidities:** Lumbar disc herniation | **Activity in natural environment** Gait Posture Index (GPI) [C] Includes scores of • Daily step count (0–25) • Step length (0–25) • Gait velocity (0–25) • Posture Upright 25 • <5 coronal or sagittal tilt 20 • 5e15 coronal or sagittal tilt 15 • >15 coronal or sagittal tilt 10 • Requiring a walking aid 5 • Unable to ambulate 0 **Equipment** Accelerometer | Baseline | **Disability** ODI **Timepoints** 3 months postoperative short | **Bivariate Analysis-Pearson correlation analysis** • Higher score on GPI at baseline is significant as prognostic factor for improvement in disability at 3 months r=0.56 P value (0.005) |

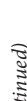

**Table 2.** (Continued)

| Characteristics of study | Objective of study | Characteristics of participants | Potential physical prognostic factors | Baseline measurement timepoint | Outcome and measure, Outcome assessment timepoints | Results |
|---|---|---|---|---|---|---|
| Gilmore et al., 2019 Australia **Design**: Prospective longitudinal study **Funding source**: Paegasus Neurosurgery Fund (Ref: PNF201502). The funding body had no role in the design of the study, collection, analysis, or interpretation of the data, or writing of the manuscript. | To investigate which variables, predict improvement of a physical function | **N = 233** **Mean age (SD)** 61(14) years **Female** n = 115(49%) **Missing Data** n = 62 (27.6%) **LBP characteristics** • Duration: <12months 45% ≥12months 49% • Distribution: Not Reported **Comorbidities**: Disc prolapse, degenerative disc disease, lumbar spinal stenosis and/or degenerative spondylolisthesis | **Activity in natural environment** • Total Walking time [C] **Equipment** ActivPAL3 accelerometer | Baseline- Over first 6 post operative days | **Oswestry Disability Questionnaire** (ODQ) **Short Form 36 Physical Component Summary** (SF-36 PCS) **Timepoints** 6 months (medium term) | **Multivariate Analysis-Regression analysis** **Disability** • Higher total walking time over the first at baseline (6 post operative days) is significant as prognostic factor for good outcome [lower disability-ODQ] at 6 months OR 1.18 95%CI (1.01–1.37) **Functional recovery** (SF-36 PCS) • Total walking time (hours) over the first 6 post operative days is not significant as prognostic factor for good outcomes [functional recovery on SF-36 (PCS)] at 6 months. Notes: Only narrative description is provided |
| Gross et al., 2005 Canada **Design** Prospective longitudinal **Funding Source** Alberta Heritage Foundation For Medical Research | To predict return to work readiness using Functional Capacity Evaluation (FCE) | **N = 130** **Mean (SD) age:** 42(10.8) Years **Female n:**67 (51%) **N = 54** (at 1 year) **Missing Data** n = 76 (58%) **LBP Characteristics** • Duration: Chronic low back pain • Distribution: Not Reported **Comorbidities**: Not Reported | **Performance based measure** • Functional capacity evaluation (FCE) [C] [FCE performance indicators (lifting, carrying, pushing, pulling, and other tasks)] • Floor to waist lift (kg) [C] **Equipment** 10kg weights | Baseline | **Return to work** Suspension of benefits Claim closure **Recurrence** By Pain disability VAS, RMDQ | **Bivariate and Multivariate Analysis – Cox Regression Analysis** **Return to work [Suspension of benefits Claim closure]** • Higher FCE performance at baseline is significant as prognostic factor for good outcome; faster return to work [faster suspension of benefits] at 12 months OR 0.91 95%CI (0.87–0.96) • Higher FCE performance at baseline is significant as prognostic factor for good outcome; faster return to work [claim closure] at 12 months OR 0.93 95%CI (0.89–0.98) • Higher weight on Floor-to-waist test at baseline is significant as prognostic factor for good outcome; faster return to work [faster suspension of benefits] at 12 months Adj: OR 1.55 95% CI (1.28–1.89) Crude: OR 1.38 95% CI (1.17–1.62) • Higher weight on Floor-to-waist test at baseline is significant as prognostic factor for good outcome; faster return to work [claim closure] at 12 months Adj: OR 1.42 95% CI (1.12–1.80) Crude: OR 1.32 95% CI (1.10–1.58) |

*(Continued)*

**Table 2.** (Continued)

| Characteristics of study | Objective of study | Characteristics of participants | Potential physical prognostic factors | Baseline measurement timepoint | Outcome and measure, Outcome assessment timepoints | Results |
|---|---|---|---|---|---|---|
| | | | | | | **Recurrence**<br>• FCE performance at baseline is not significant as prognostic factor for good outcome; lower rate of recurrence [disability-RMDQ] at 12 months<br>P value (0.07)<br>• FCE performance at baseline is not significant as prognostic factor for good outcome; lower rate of recurrence [pain intensity-VAS] at 12 months<br>P value (0.09)<br>• Floor to waist lift at baseline is not significant as prognostic factor for good outcome; lower rate of recurrence [disability] at 1 year<br>P value (0.26)<br>Note: recovery expectations and workplace support were controlled(adj) |
| Grotle et al., 2007<br>Norway<br>**Design**<br>Prospective longitudinal study<br>**Funding Source**<br>Norwegian fund for Post Graduate Training in Physiotherpay and Norwegian Back Pain Network | To examine the clinical course of acute low back pain and to evaluate prognostic factors for recovery. | **N** = 123<br>**Mean age (SD)** 38.9 (10.3) years<br>**Female 71** (58%)<br>**Missing Data** n = 3 (2%)<br>**LBP Characteristics**<br>• Duration: Less than 3 weeks<br>• Distribution: With or without radiating pain<br>**Comorbidities**: Not Reported | **Impairment-based measure**<br>• Fingertip to floor distance (cm) [C]<br>**Equipment**<br>Tape measure | Baseline | **Pain intensity** (NRS)<br>**Timepoints**<br>3 months (short term) | **Multivariate Analysis-Logistic Regression Analysis**<br>**Non- Recovery**<br>• Finger tip to floor distance at baseline is not significant as prognostic factor for non recovery at 3 months<br>OR P value (> 0.10) 95% CI – 0.60 (0.18–2)<br>Note: Adjusted for age and gender |

*(Continued)*

**Table 2.** (Continued)

| Characteristics of study | Objective of study | Characteristics of participants | Potential physical prognostic factors | Baseline measurement timepoint | Outcome and measure, Outcome assessment timepoints | Results |
|---|---|---|---|---|---|---|
| Gurcay et al., 2009 Turkey **Design** Prospective longitudinal study **Funding Source** Not Reported | To assess the clinical course of patients with acute low back pain (LBP) throughout 12 weeks and to identify the prognostic factors for non-recovery in the short term. | **N = 99** **Mean age (SD)** 37.9 (3.2) years **Female 33** (36%) **Missing Data** n = 8 (8.7%) **LBP Characteristics** • Duration: Less than 3 weeks • Distribution: Not Reported **Comorbidities:** Not Reported | **Impairment-based measure** • Fingertip to floor distance (cm) [C] **Equipment** Tape measure | Baseline | **Recovery** Recovery was considered if patients scored <4 on the RMDQ and pain had resolved **Timepoints** 2 weeks, 12 weeks (short term) | **Bivariate Analysis-Mann–Whitney U test** **Recovery** • Smaller finger to floor distance at baseline is significant as prognostic factor for early recovery at 2 weeks P value (0.005), P<0.25 is significance level • Finger to floor distance is not significant as prognostic factor for non recovery at 12 weeks Note: It was not included in Regression Analysis |
| Hazard et al., 1991 Denmark **Design:** Prospective Longitudinal Study **Funding source** Unfunded | To investigate disability exaggeration, predict which patients will return to work | **N = 258** **Mean age (SD)** 11.5 (2.2) Years **Female** 32.9% **LBP characteristics** • Duration of pain mean 14.7 months • Distribution: Not Reported **Comorbidities:** Not Reported | **Impaired based measures** • Trunk flexibilities (flexion extension ROM) [C] **Performance based measures** • Frequent Floor to waist lifting (kg) [C] • Cycling endurance [C] **Equipment** Inclinometer 2.27 kg weights Stationary bicycle | Baseline | **Work status** Return to work **Timepoints** 1 year 2 years (Long term) | **Bivariate Analysis-Chi Square-Analysis** **Work status (Return to work)** • Trunk flexibility (flexion plus extension ROM) at baseline is not significant as prognostic factor for work status (return to work) at 1 year P value>0.05 • Floor to waist lifting at baseline is not significant as prognostic factor for work status (return to work) at 1 year P value>0.05 • Cycling endurance at baseline is significant as prognostic factor for better outcome, [return to work] at 1 year P value< 0.05 • Trunk flexibility (flexion plus extension) at baseline is not significant as prognostic factor for work status (return to work) at 2 years P value>0.05 • Floor to waist lifting at baseline is not significant as prognostic factor for work status (return to work) at 2 years P value>0.05 • Cycling endurance at baseline is not significant as prognostic factor for work status at 2 years. P value>0.05 |

*(Continued)*

**Table 2.** (Continued)

| Characteristics of study | Objective of study | Characteristics of participants | Potential physical prognostic factors | Baseline measurement timepoint | Outcome and measure, Outcome assessment timepoints | Results |
|---|---|---|---|---|---|---|
| Hendrick et al., 2013 UK **Design** Prospective longitudinal study **Funding Source** University of Otago establishment | To assess the predictive relationship between activity and disability at 3 months in a sub-acute LBP population. | **N = 101** **Mean (SD) age** 37.8(14.6) years **Female:**51 (50.5%) **Missing data** n = 18 **LBP Characteristics** • Duration: 6 weeks or less • Distribution: Not Reported **Comorbidities** No pre-existing conditions that may limit mobility | **Measure of activity in natural environment** **Physical activity [C]** • RT3 VM/hr/wk • PAEE kcals/kg • Change in RT3 VM/r/wk from baseline to 3 months • RT3VM_change in low RT3 VM/hr group at baseline • Change in PAEE kcals/kg from baseline to 3 months • PAEE kcals/kg change (7D-PAR) baseline **Equipment** RT3 accelerometer | Baseline | **Disability** RMDQ RMDQ change **Timepoints** 3 months (Short term) | **Multivariate Analysis-Linear regression analysis** **Univariate analysis** • RT3 VM/hr/wk at baseline is not significant as prognostic factor for disability [RMDQ] at 3 months B 0.000 P value (0.20) 95%CI (0.000–000) • PAEE kcals/kg at baseline is not significant as prognostic factor for disability [RMDQ] at 3 months B 0.005 P value (0.59) 95%CI (−0.15–0.026) • Change in RT3 VM/r/wk from baseline to 3 months is not significant as prognostic factor for disability [RMDQ] at 3 months B 0.00 P value (0.33) 95%CI (0.00–0.00) • RT3VM_change in low RT3 VM/hr group at baseline months is not significant as prognostic factor for disability [RMDQ] at 3 months B 0.004 P value (0.62) 95% CI (0.006–0.000) • Change in PAEE kcals/kg from baseline to 3 months at baseline months is not significant as prognostic factor for disability [RMDQ] at 3 months B 0.046 P value (0.52) 95%CI (−0.096–0.188) • PAEE kcals/kg change (7D-PAR) in low activity group at baseline is not significant as prognostic factor for disability [RMDQ] at 3 months B 0.009 P value (0.63) 95%CI (−0.27–0.05) **Multiple linear regression analysis** • RT3 VM/hr/wk change at 3 months is not significant as prognostic factor for disability change [RMDQ change] at 3 months B 0.00 P value (0.81) 95% CI (0.00–0.00) • RT3VM change at 3 months is not significant as prognostic factor for change in disability [RMDQ change] at 3 months B 0.00 P value (0.89) 95%CI (0.00–0.00) PAEE kcals/kg change at 3 months is not significant as prognostic factor for disability change [RMDQ change] at 3 months B −0.001 P value (0.45) 95%CI (−0.41–0.02) |

*(Continued)*

**Table 2.** (Continued)

| Characteristics of study | Objective of study | Characteristics of participants | Potential physical prognostic factors | Baseline measurement timepoint | Outcome and measure, Outcome assessment timepoints | Results |
|---|---|---|---|---|---|---|
| Hicks, et al. 2005 US **Design** Prospective longitudinal study **Funding Source** Foundation for Physical Therapy Clinical Research Center | To develop a clinical prediction rule to predict treatment response to a stabilization exercise program for patients with low back pain | **N = 54** **Mean (SD) age** 42.4 (12.7) years **Female:** 57.4 **Missing Data** n = 3 **LBP Characteristics** • Duration: 40.6 (44.2 days) • Distribution: Back/buttock only (%) 53.7 Distal symptoms (%) 46.3 Prior history LBP (%) 70.4 **Comorbidities:** Not Reported | **Impairment based measures** • Average SLR- ROM [C] • Lumbar flexion ROM [C] **Equipment** Single Inclinometer | Baseline | **Disability** ODQ **Timepoints** 8 weeks (short term) | **Bivariate Analysis** **Disability (ODQ)** • Higher average SLR-ROM at baseline is significant as prognostic factor for good outcome; lower disability [ODQ] at 8 weeks Positive LR 3.3 (0.90–12.4) P value (0.069) Level of sig (P 0.10) • Lower lumbar flexion ROM at baseline is a significant prognostic factor for poor outcome; higher disability [ODQ] at 8 weeks Positive LR1.3 P value (0.058) Level of sig (P 0.10) **Multivariate Analysis-Step wise Regression Analysis** • Higher SLR ROM at baseline is significant as a prognostic factor for good outcomes; reduced disability [ODQ] at 8 weeks (Narrative description is provided only) • Lumbar flexion ROM at baseline is not significant prognostic factor for outcomes; disability [ODQ] at 8 weeks (Narrative description is provided only) |
| Hildebrandt et al. 1997 Germany **Design** Prospective longitudinal study **Funding source:** Not Reported | To determine whether objective or subjective signs most influence the outcome of rehabilitation | **N = 90** **Mean age** 42 years **Female** n = 21 (51%) **Missing Data** = 8 (9%) **LBP characteristics** • Duration: Chronic LBP • Distribution: Other location of pain 63% Radicular pain 26% **Comorbidities** Disabled Depression = 68% Nonspecific bodily pain = 48% | **Impairment based measures** • Finger to floor distance (cm) [C] **Performance Based measures** • Trunk flexion performance (reps) • Trunk extension performance test (reps) **Equipment** Tape measure | Baseline | **Return to work** **Pain Reduction** **Patients own rating of success** 4 Point Likert scale **Timepoints** 8 weeks | **Bivariate Analysis- Chi square test** **Return to work** • Higher trunk flexion performance is significant as prognostic factor at baseline for good outcome [back to work] at 8 weeks X² 4.7 P value (< 0.05) • Lower finger to floor distance is significant as prognostic factor at baseline for good outcome [back to work] at 8 weeks X² 5.4 P value (< 0.05) **Reduction of Pain Intensity** • Higher trunk extension test (rep) is not significant as prognostic factor at baseline for good outcome [reduction of pain intensity] at 8 weeks. X² 6.3 P value (< 0.05) **Patients own rating of success** • Lower finger to floor distance is significant as prognostic factor at baseline for good outcome [Patients own rating of success] at 8 weeks X² 3.9 P value (< 0.05) • Higher trunk extension performance (ROM) is significant as prognostic factor at baseline for good outcome [patients own rating of success] at 8 weeks X² 5.4 P value (<0.05) |

*(Continued)*

| Characteristics of study | Objective of study | Characteristics of participants | Potential physical prognostic factors | Baseline measurement timepoint | Outcome and measure, Outcome assessment timepoints | Results |
|---|---|---|---|---|---|---|
| Hirayama et al. 2019 Japan **Design** Prospective longitudinal study **Funding Source** Not Reported | To develop a clinical prediction rule (CPR) that predicts treatment responses to mechanical lumbar traction (MLT) among patients with lumbar disc herniation (LDH | **N = 103** **Mean (SD)age** 43.7(14.1) years **Female:** 43 (41%) **Missing Data = 24** (19%) **LBP Characteristics** • Duration: Not Reported • Distribution: Low back symptoms only (%) (Responder)37.5% (Non responders)21.5% Buttock/thigh symptoms present (%) R) 50.0% (Not Reported)68.4% Symptoms distal to knee (%) (R) 45.8% (Not Reported) 40.5% **Comorbidities:** Not Reported | **Impairment- based measure** • Lumbar flexion ROM [C] Lumbar extension ROM [C] **Equipment** Tape measure | Baseline Pre-treatment | **Disability** ODI **Timepoints** 2 weeks (short term) | **Bivariate Analysis-U test** **Disability (ODI)** • Lumbar flexion ROM at baseline is not significant as a prognostic factor for disability [ODI] at 2 weeks P value (0.285) for U test **Multivariate Analysis – Logistic regression** • Higher lumbar extension at baseline is significant as prognostic factor for higher disability [ODI] at 2 weeks OR 5.39 P value (0.005) 95%CI (1.65–17.66) Level of sig (0.15) |
| Jain et al. 2023 India **Design:** Prospective longitudinal Study **Funding source** Unfunded | To determine the correlation between post-treatment trunk range of motion (ROM) and isometric strength (TIS) and pain and disability in patients who underwent multimodal rehabilitation for low back pain (LBP). | **N = 266** **Mean age (SD) 45.6** (15.2) years **Female: 54** (44%) **Missing Data** n = 144 **LBP Characteristics** • Duration: Not Reported • Distribution: low back and leg pain **Comorbidities:** Not Reported | **Impairment-based measures** • Isometric Trunk muscle strength (flexion, extension) [C] • Trunk extension ROM [C] • Trunk flexion ROM [C] **Equipment** Dynamometer | Baseline | **Pain intensity** NPRS **Disability** ODI **Timepoints** Post treatment After 6 sessions Short-term | **Bivariate Analysis** **Pain Intensity** • Higher extension ROM at baseline is significant as prognostic factor for lower pain intensity after 6 sessions r=−0.24, P value (0.006) • Higher flexion ROM at baseline is significant as prognostic factor for lower pain intensity after 6 sessions r=−0.28, P value (0.001) **Disability** • Higher extension strength at baseline is significant as prognostic factor for decreased disability after 6 sessions r=−0.30, P value (0.0007) • Higher flexion strength at baseline is significant as prognostic factor for decreased disability after 6 sessions r=−0.28, p value (0.001) Note: All other factors are non significant for pain and disability |

*(Continued)*

| Characteristics of study | Objective of study | Characteristics of participants | Potential physical prognostic factors | Baseline measurement timepoint | Outcome and measure, Outcome assessment timepoints | Results |
|---|---|---|---|---|---|---|
| Jakobsson et al. 2019 Sweden **Design** Prospective longitudinal study **Funding Source** AFA Research Funding, Eurospine Research Grants, the Health and Medical Care Executive Board of the Västra Götaland Region, Doctor Felix Neubergh grants, and Renée Eander's Help Fund. | To investigate the predictive value of preoperative fear-avoidance factors (self-efficacy for exercise, pain catastrophizing, kinesiophobia, and depression), walking capacity, and traditional predictor variables for predicting postoperative changes in physical activity level and disability 6 months after lumbar fusion surgery in patients with chronic low back pain (LBP). | **N = 118** **Mean (SD) age** 46.4 (8.2) years **Female:** 49 (54.4) **Missing data** n=28 (23%) **LBP Characteristics** • Duration: Chronic • Distribution: with or without leg pain **Comorbidities** Lumbar fusion surgery, could have additional minor radiating symptoms with or without a simultaneous surgical procedure for disc herniation, isthmic spondylolisthesis, or foraminal stenosis | **Performance-based measure** • Walking capacity by 5 min walk (distance covered) [C] • 15 m distance covered (time) [C] **Activity in natural environment** • Preoperative physical activity [C] **Equipment** Accelerometer | Baseline | **Physical activity level** Triaxial accelerometer GT3X+ **Disability** ODI **Timepoints** 6 months (medium term) | **Multivariate Analysis-Multiple Linear Regression Model** **Physical Activity** • Higher Preoperative physical activity level at baseline is significant as prognostic factor lower higher change in physical activity level at 6 months B −0.349 P (< 0.001) 95% CI (-.0.482- −0.216) R2 0.251 **Disability** • Higher preoperative disability at baseline is significant as prognostic factor for lower change in disability from baseline to 6 months B −0.790 P value (< 0.001) 95%CI (−1.026- −0.553) **Univariate Analysis** **Physical Activity** • Walking capacity (5 min walk test) at baseline is not significant as prognostic factor for change in physical activity at 6 months P (> 0.25) (NS>0.25) • Walking capacity (15-meter walk test) at baseline is not significant as prognostic factor for change in physical activity at 6 months P (> 0.25) **Disability** • Walking capacity (5 min walk test) at baseline is not significant as prognostic factor for change in disability at 6 months P (> 0.25) (NS>0.25) • Walking capacity (15-meter walk test) at baseline is not significant as prognostic factor for change in disability at 6 months P (> 0.25) |

Table 2. (Continued)

| Characteristics of study | Objective of study | Characteristics of participants | Potential physical prognostic factors | Baseline measurement timepoint | Outcome and measure, Outcome assessment timepoints | Results |
|---|---|---|---|---|---|---|
| Karp et al. 2015 USA **Design** Prospective longitudinal study **Funding** NIH grant | To estimate relationship between presurgical variables and outcomes of great importance to patients-back related disability and satisfaction with surgery | **N = 48** **Mean (SD) age:** 71.4 ± 8.0 years **Female n (%):** 22 (45.8%) **Missing data:** 7 **LBP characteristics** • Duration: Not Reported • Distribution: LBP with or without radiculopathy **Comorbidities:** Spinal stenosis | **Performance based measure** • 4-meter walk test (Gait Speed) [C] **Equipment** Stopwatch | Baseline | **Disability** RMDQ **Satisfaction with treatment** SSSQ **Timepoints** 3 months (Short term) | **Bivariate Analysis-Pearson Correlations Disability (RMDQ)** • 4-meter gait speed at baseline is not significant as prognostic factor for disability at 3 months r = 0.004 P value (0.98) **Satisfaction (SSSQ)** • 4-meter gait speed at baseline is not significant as prognostic factor for patients' satisfaction with treatment at 3 months r = 0.119 P value (0.46) |
| Kool et al. 2002 Switzerland **Design** Prospective longitudinal study **Funding source** Grant from Klinik Valens | To determine the predictive ability of a modified version of the Step Test, the Pseudo Strength Test and a pain intensity of 9 or 10 NRS for non-RTW in patients with CLBP. | **N = 99** **Mean (SD) age** 41.8 (8.9) years **Female** 15 (15%) **Missing Data n = 8** (9%) **LBP Characteristics** • Duration: CLBP • Distribution: with or without radiculopathy **Comorbidities:** Not Reported | **Performance based measure** • Step Test [D] • Pseudo Strength Test [C] **Equipment** 30 cm step 3 Kg weights | Baseline | **Non- Return to Work (RTW)** **Timepoints** 12 months (Long term) | **Bivariate Analysis-Logistic Regression Analysis** **Non-Return to work** • Positive Step test (Precipitous cessation is counted as a positive test) is significant as prognostic factor at baseline for poor outcome [non return to work] at 12 months χ2 0.0050 PPV (0.95) • Low Pseudo Strength is significant as prognostic factor at baseline for non return to work at 12 months χ2 0.0470 PPV (0.93) **Multivariate Analysis** • A combination of two or tests showed higher predictive accuracy for non-return to work PPV (0.97) |
| Lagersted-Olsen eta al. 2016 Denmark **Design** Prospective longitudinal study **Funding Source** The National Research Centre for the Working Environment (Not Reported CWE) | To investigate high levels of FBW increases the risk for aggravation of LBP among workers reporting LBP at baseline (LBP intensity >0). | **N = 482** **Mean age 46** **Female 56%** **Missing Data = Not Reported** **LBP Characteristics:** Not Reported **Comorbidities:** Not Reported | **Activity in natural environment** • Forward bending FBW>60° (minutes/day) [D] **Equipment** Accelerometer (Acti-Graph GT3X) | Baseline | **Pain intensity** Modified version of the standardized Nordic Questionnaire for the Analysis of Musculoskeletal Symptoms **Timepoints** Every month for 12 months Long term | **Bivariate Analysis** **Pain Intensity** • FBW>60 deg at baseline is not significant as a prognostic factor for good outcome [reduced pain intensity-Nordic questionnaire] at 12 months HR 0.18 95%CI (0.20–0.56) **Multivariate Analysis** • FBW>60°at baseline is not significant as prognostic factor for good outcome [reduced pain intensity- Nordic questionnaire] at 12 months HR 0.14 95%CI (−0.17–0.46) |

*(Continued)*

**Table 2.** (Continued)

| Characteristics of study | Objective of study | Characteristics of participants | Potential physical prognostic factors | Baseline measurement timepoint | Outcome and measure, Outcome assessment timepoints | Results |
|---|---|---|---|---|---|---|
| Lee et al. 2017 USA **Design** Prospective longitudinal study **Funding source** Zion Charity Foundation | To identify predictors for postoperative clinical outcome in lumbar stenosis using smart shoe technology | **N = 29** **Mean age (SD)** 59.1 (15.9) years **Female n = 21** **Missing Data = 14** **LBP characteristics** • Duration: Not Reported • Distribution: Radiculopathy or axial pain in lower limb **Comorbidities** Lumbar disk herniation, Lumbar spondylolisthesis, and/or adjacent segment disease, lumbar spinal stenosis | **Performance- based measures** 10m self paced walking test (SPWT) [C] • StdTime-P2P3-max • CrossCorr-P2 • AutoCorr-P2-Mean • SumMag-P2-Min • MeanTime-p2P3-Mean **Equipment** Sensorized smart shoes | Preoperative | **Disability** ODI **Timepoints** 3 months Postoperatively (short term) | **Bivariate Analysis- Spearman correlation analysis** **Disability [ODI]** • High StdTime-P2P3 _Max preoperatively is significant as prognostic factor for good outcome [low disability; ODI] at 3 months postoperatively r 0.61 P value (0.016) • High CrossCorr-P2 preoperatively is significant as prognostic factor for good outcome [low disability: ODI] at 3 months postoperatively r 0.54 P value (0.037) • Higher autocorrp2-mean preoperatively is significant as prognostic factor for good outcome [low disability: ODI] at 3 months postoperatively r 0.54 P value (0.043) • Higher SumMag-P2-Min-P2_min preoperatively is significant as prognostic factor for good outcome low disability; ODI] at 3 months postoperatively r 0.51 P value (0.053) • Mean Time-P2P3-Mean preoperatively is not significant as prognostic factor for good outcome [low disability; ODI] at 3 months postoperatively r 0.49 P value (0.065) |
| Lubetzky et al. 2019 USA **Design** Prospective longitudinal study **Funding Source** Unfunded | To determine whether balance at baseline predicts long-term postsurgical outcomes. | **N = 43** **Mean age (SD)** 62.7 (10.8) **Missing Data = 0** **LBP Characteristics** • Duration: CLBP • Distribution: With/ without leg pain **Comorbidities** Degenerative changes of lumbar spine, Lumbar Spine surgery (lumbar fusion, microdiscectomy, decompression) | **Performance- based measure** • Single-leg stance (SLS) test [C] • Four square step test (FSST) [C] • 8 Foot up and go test (TUG) [C] **Equipment** Stop watch | Baseline | **Disability** ODI **Timepoints** 12 months (Long term) | **Multivariate Analysis-Linear mixed effect model** **Disability** • Higher performance of three tests single stance test, 4 square step test, 8 up and Go test was significant as prognostic factor for good outcomes [reduced disability-ODI] at 12 months R2 = 0.36 |

*(Continued)*

**Table 2.** (Continued)

| Characteristics of study | Objective of study | Characteristics of participants | Potential physical prognostic factors | Baseline measurement timepoint | Outcome and measure, Outcome assessment timepoints | Results |
|---|---|---|---|---|---|---|
| Mellin et al. 1993 Finland **Design:** Prospective longitudinal study **Funding Source** Not Reported | To determine whether physical measurements predict outcome after multimodal treatment including intensive physical training of patients with CLBP | **N = 194** **Female** 97 (56%) **Mean (SD) age** 42.8 (7.2) years **Missing Data =** Not Reported **LBP characteristics** • Duration: Chronic • Distribution: Not Reported **Comorbidities:** Not Reported | **Impairment-based measures** • Spinal mobility (Total of all tests) [C] • Trunk flexion strength [C] • Trunk extension strength [C] **Performance-Based measure** • Isokinetic Lifting strength **Equipment** Myrin Inclinometer Spring balance dynamometer Digitest OY | Baseline | **Functional Capacity Index (FCI)** **Return to work** **Timepoints** 12 months (Long term) | **Multivariate Analysis-Stepwise multiple and Logistic regression** **Functional capacity** • Higher spinal mobility from baseline to 12 months is significant as prognostic factor for improved FCI in women B 0.29 P (< 0.05) Note: (It was non significant in men) **Return-to-work** • Higher spinal mobility from baseline to 12 months is significant as prognostic factor for earlier return to work OR 1.06 P value (< 0.01) 95% CI (1.00–1.11) Note: Only narrative description is provided for other tests: Trunk flexion strength, extension flexion strength and Isokinetic Lifting strength is not significant as prognostic factors for FCI and return to work |
| Milhouse et al. 1989 Canada **Design:** Prospective longitudinal study **Funding source** Grant from the Social Security Administration | To determine whether specific physical or psychologic findings could predict return to work in a group of patients admitted to an orthopedic back pain clinic in an effort to clarify the extent to which physical impairment evaluations assess vocational disability | **N = 87** **Mean age 39.5 Years** SD: Not Reported **Female** (Not Reported) **Missing Data** (Not Reported) **LBP Characteristics** • Duration: Not Reported • Distribution: Leg pain60 (31.7%) **Comorbidities:** Not Reported | **Impairment-based measure** • Isometric trunk extension strength [C] • Trunk ROM [C] **Performance-based measure** • Ability to lift-frequent lifting [C] **Equipment** 2 sagittal plane coordinate measuring devices fixed to a vertical frame | Baseline | **Return to Work** **Timepoints** 6 months (medium term) | **Bivariate Analysis- Chi-square test** **Return to Work** • Trunk strength, ROM and ability to lift are not significant as prognostic factors at baseline for good outcome [return to work] at 6 months P (> 0.05) Notes: Only narrative description is provided |

*(Continued)*

**Table 2.** (Continued)

| Characteristics of study | Objective of study | Characteristics of participants | Potential physical prognostic factors | Baseline measurement timepoint | Outcome and measure, Outcome assessment timepoints | Results |
|---|---|---|---|---|---|---|
| Moradi et al. 2009<br>Germany<br>**Design**<br>Prospective longitudinal study<br>**Funding Source**<br>Unfunded | To evaluate the value of three commonly used physical performance tests, the Biering–Sørensen, Oesch and Villiger tests, for predicting the success of multidisciplinary pain treatment in patients with LBP | **N = 162**<br>**Missing:** Not Reported<br>**Mean (SD) age** 46 (11) years<br>**Female** 76(46%)<br>**LBP characteristics**<br>• Duration: At least 6 weeks'<br>• Distribution: Not Reported<br>**Comorbidities:** Not Reported | **Performance-based measures**<br>• Villiger test [C]<br>• Oesch test [C]<br>• Biering–Sørensen test [C]<br>**Equipment**<br>30cm step<br>3kg weights | Baseline (before treatment) | **Pain intensity** (VAS)<br>**Disability** (PDI)<br>**Functional back capacity** (FFbH-R)<br>**Timepoints**<br>3 weeks<br>6 months<br>(Short Medium term) | **Bivariate Analysis- Pearson correlation coefficients**<br>**Pain Intensity (VAS)**<br>• Higher performance of villager test at baseline is significant as prognostic factor for reduced pain [VAS] at 3 weeks<br>r −0.43 P value (<0.001)<br>• Higher performance of villager test at baseline, is not significant as prognostic factor for reduced pain [VAS] at 6 months<br>r −0.32 P value (>0.05)<br>• Higher performance of oesch test at baseline is significant as prognostic factor for reduced pain [VAS] at 3 weeks<br>r −0.34 P value (<0.001)<br>• Higher performance of oesch test at baseline is significant as prognostic factor for reduced pain [VAS] at 6 months<br>r -0.26 P value (<0.001)<br>• Higher performance of Biering–Sørensen test (extension strength) at baseline is significant as prognostic factor for reduced pain [VAS] at 3 weeks<br>r-0.28 P value (<0.001)<br>• Higher performance of Biering–Sørensen test (extension strength) at baseline is significant as prognostic factor for reduced pain [VAS] at 6 months<br>r-0.38 P value (<0.001)<br>**Disability (PDI)**<br>• Higher performance of villager test at baseline, is significant as a prognostic factor for reduced disability [PDI] at 3 weeks<br>r-0.37 P value (<0.001)<br>• Higher performance of villager test at baseline, is not significant as prognostic factor for reduced disability [PDI] at 6 months<br>r-0.34 P value (>0.001)<br>• Higher performance of oesch test at baseline is significant as prognostic factor for reduced disability [PDI] at 3 weeks<br>r −0.40 P value (<0.001)<br>• Higher performance of oesch test at baseline is significant as prognostic factor for reduced disability [PDI] at 6 months<br>r-.0.30 P value (<0.001) |

*(Continued)*

**Table 2.** (Continued)

| Characteristics of study | Objective of study | Characteristics of participants | Potential physical prognostic factors | Baseline measurement timepoint | Outcome and measure, Outcome assessment timepoints | Results |
|---|---|---|---|---|---|---|
| | | | | | | • Higher performance of Biering–Sørensen test (extension strength) at baseline is significant as prognostic factor for reduced disability [PDI] at 3 weeks<br>r-.0.25 P value (<0.001)<br>• Higher performance of Biering–Sørensen test (extension strength) at baseline is significant as prognostic factor for reduced disability [PDI] at 6 months<br>r-.0.30 P value (<0.001)<br>**Functional back capacity (FFbH-R)**<br>• Higher performance of villager test at baseline is significant as a prognostic factor for higher functional capacity (FFbH-R) at 3 weeks<br>r 0.41 P value (<0.001)<br>• Higher performance of villager test at baseline is not significant as prognostic factor for functional capacity [FFbH-R] at 6 months<br>r-0.54 P value (>0.001)<br>• Higher performance of oesch test at baseline is significant as prognostic factor for higher functional capacity [FFbH-R] at 3 weeks<br>r −0.33 P value (<0.001)<br>• Higher performance of oesch test at baseline is significant as prognostic factor for higher functional capacity [FFbH-R] at 6 months<br>r-.0.36 P value (<0.001)<br>• Higher performance of Biering–Sørensen test (extension strength) at baseline is significant as prognostic factor for improved functional capacity (FFbH-R) at 3 weeks<br>r-.0.40 P value (<0.001)<br>• Higher performance of Biering–Sørensen test (extension strength) at baseline is significant as prognostic factor for improved functional capacity (FFbH-R) at 6 months<br>• r-.0.53 P value (<0.001) |

*(Continued)*

**Table 2.** (Continued)

| Characteristics of study | Objective of study | Characteristics of participants | Potential physical prognostic factors | Baseline measurement timepoint | Outcome and measure, Outcome assessment timepoints | Results |
|---|---|---|---|---|---|---|
| Nordeman et al. 2017,2014 Sweden **Design:** Prospective Longitudinal Study **Funding Source** Research and development council of Södra Älvsborg, Region Västra Götaland, Sweden. The Health & Medical Care Committee of the Regional Executive Board, Region Västra Götaland, Sweden | (2017) To investigate prognostic factors for future activity limitation in women with chronic low back pain (CLBP) consulting primary health care (2014) To investigate prognostic factors for future work ability in women with chronic low back pain (CLBP) consulting primary health care | **N = 130** **Means age (SD)** 45(10) Years **Female** 130 (100%) **Missing Data =** 7 (5%) **LBP Characteristics** • Duration: LBP duration >12-week • Distribution: with or without referred leg pain. **Comorbidities:** Not Reported | **Impairment based measure** • Hand grip strength [C] **Performance based measure** • 6 min walk Distance (6MWD) [C] **Equipment** Grippit | Baseline | **Activity limitation** (RMDQ) **Self reported work ability**- (yes/ No) **Timepoints** 2 years (Long term) | **Bivariate Analysis -Pearson Correlation test (2017)** **Activity Limitation (RMDQ)** • Handgrip strength at baseline is not significant as prognostic factor for activity limitation at 2 years r −0.17 P value (0.070) • Higher 6MWT at baseline is significant as prognostic factor for reduced activity limitation at 2 years r −0.41 P value (< 0.0001) **Multivariate analysis** • Hand grip strength at baseline is not significant as prognostic factor for activity limitation at 2 years OR0.057 P-value (0.55) Level of sig p<0.20 • Higher 6MWT at baseline is significant as prognostic factor for reduced activity limitation at 2 years. OR −0.23 P value (0.020) 95% CI (−0.42–0.036) **Univariate Analysis- Logistic Regression (2014)** **Work ability (Yes/No) as outcome** • Higher 6MWT at baseline is significant as prognostic factor for improved work ability at 2 years OR 3.7 P value (< 0.0001) 95% CI (2.0–6.8) • Higher Hand grip strength at baseline is significant as prognostic factor for improved work ability at 2 years. OR 1.8 P value (< 0.0001) 95% CI (1.3–2.5) **Multivariate Analysis** • Higher 6MWT at baseline was significant as prognostic factor for improved work ability at 2 years OR 3.3 P value (< 0.0036) 95%CI (1.5–7.4) |

*(Continued)*

**Table 2.** (Continued)

| Characteristics of study | Objective of study | Characteristics of participants | Potential physical prognostic factors | Baseline measurement timepoint | Outcome and measure, Outcome assessment timepoints | Results |
|---|---|---|---|---|---|---|
| Patterson et al., 2022 Sydney **Design** Prospective longitudinal study **Funding Source** This research received competitive funding from the University of Sydney Bridging Support Grant—NHMRC | The aim of this study was to assess the relationship between different domains (e.g., leisure, transport, household, work-related), and intensities (e.g., moderate, vigorous) of physical activity assessed via the device and self-reported questionnaires, and the frequency of analgesic use and activity limitation in people with LBP | **N = 160** **Mean (SD) age** 56.1(5.1) years **Female:** 118 (74%) **Missing data** n=28 (17%) **LBP Characteristics:** Not Reported **Comorbidities:** Not Reported | **Activity in natural environment** • Intensity of Physical activity Moderate-vigorous physical activity (min/week) [C] • Physical workload (score 0–62) • Sedentary behaviour time (min/week) [C] • Domain of Physical activity (MET min/week) [C] Leisure physical activity Transport physical activity Household physical activity Work physical activity **Equipment** Accelerometers | Baseline | **Analgesic Use** Use Frequency **Activity Limitation** Number of activity limitation counts **Timepoints** 12 months | **Multivariate Analysis-Binomial Model Analysis** **Analgesic Use** • Higher time spent in moderate-vigorous physical activity at baseline is significant as prognostic factor for lower analgesic use counts at 1 year IRR 0.97 95%CI (0.96–0.99) • Higher physical workload at baseline is significant as prognostic factor for higher analgesic use counts at 1 year IRR1.02 95%CI (1.01–1.05) • Sedentary time at baseline is not significant as prognostic factor for analgesic use counts at 1 year IRR 1.06 P value (0.07) 95 CI% (0.93–1.11) • Physical activity domains (leisure, transport, household, work PA) at baseline are not significant as prognostic factors for analgesic use counts at 1 year. Leisure PA IRR 0.69 P value (0.07) 95% CI (0.55–1.08) Transport PA IRR 0.83 P value (0.08) 95%CI (0.79–0.89) Household PA IRR 1.04 P value (0.07) 95% CI (0.98–1.13) Work PA IRR 1.06 P value (0.06) 95% CI (0.92–1.63) **Activity Limitation** • Higher sedentary time at baseline is significantly associated with increased number of activity limitation counts at 1 year IRR1.04 P value (0.04) 95% CI (1.01–1.09) • Moderate-vigorous physical activity at baseline is not significant as prognostic factor for activity limitation counts at 1 year IRR 0.91 P value (0.11) 95%CI (0.88–1.21) • Physical workload at baseline is not significant as prognostic factor for activity limitation counts at 1 year IRR 1.20 P value (0.07) 95%CI (0.93–1.44) • Domains of physical activity (leisure, transport, household, work PA) at baseline is not significant as prognostic factors for activity limitation counts at 1 year |

*(Continued)*

| Characteristics of study | Objective of study | Characteristics of participants | Potential physical prognostic factors | Baseline measurement timepoint | Outcome and measure, Outcome assessment timepoints | Results |
|---|---|---|---|---|---|---|
| | | | | | | Leisure physical activity IRR 0.94 P value (0.05) 95%CI (0.81–0.99) Transport physical activity IRR 0.93 P value (0.06) 95% CI (0.88–1.03) Household physical activity IRR 1.01 P value (0.09) 95%CI (0.95–1.22) Work physical activity IRR 1.01 P value (0.10) 95% CI (0.93–1.33) |
| Rodriguez-Romero et al. 2022 Switzerland **Design** Prospective longitudinal study **Funding** No funded | To determine: (i) the effect of time spent standing on pain status during a 1-h laboratory-based standing task; (ii) the point after which significant increases in pain are likely; and (iii) the individual (e.g., age, sex, history of LBP, self-rated health), physical (e.g., deficits in motor control, muscle endurance) and psycho-social (e.g., job demands) factors that are associated with higher levels of low-back and lower limb pain after a 1-h standing task | **N = 40** **Mean (SD) age:** 37.4 ± 6.6 **Female n (%):**55% **Missing data:** 0 **LBP characteristics:** • Duration: Not Reported • Distribution: low back and lower extremities **Comorbidities:** Not Reported | **Performance-based measures** • Higher hip abductor muscle endurance (sec) [C] • Isometric Hip Abduction endur-ance test [C] • Active hip abduction test (AHAbd) [C] • Active straight leg raise test (ASLR) [C] • Supine bridge [C] • Biering–Sorensen test [C] **Equipment** Stop watch | Baseline | **Pain status** (Yes/no) **Severity of pain** (VAS, 0–100 mm) **Timepoints** 15, 30,45,60 min (Short term) | **Multivariate Linear Regression Model** **Pain intensity** • Higher hip abductor muscle endurance at baseline is significant as prognostic factor for lower low back pain intensity throughout 1-hour standing task. B −0.233, SE0.081, P value (0.007), 95%CI (−0.397 – −0.069) **Mixed regression analysis:** • Higher time on hip abductor muscle endurance test at baseline is significant as prognostic factor for lower low back pain intensity throughout 1-hour standing task B −0.004, SE 0.002, P value (0.022) **Univariate Analysis-Spearman Correlation** • Biering–Sorensen test at baseline is not sig-nificant as prognostic factor for low back pain intensity throughout 1 -hour standing task P value (0.070) (NS > 0.05) • Supine bridge at baseline is not significant as prognostic factor for low back pain intensity through 1-hour standing task. P value (0.062) • Active straight leg raise test (ASLR) at baseline is not significant prognostic factor for low back pain intensity throughout 1-hout standing task P value (0.193) • Active hip abduction test (AHAbd) at baseline is not significant as prognostic factor for low back pain intensity through 1-hour standing task P value (0.0.816) |

*(Continued)*

| Characteristics of study | Objective of study | Characteristics of participants | Potential physical prognostic factors | Baseline measurement timepoint | Outcome and measure, Outcome assessment timepoints | Results |
|---|---|---|---|---|---|---|
| Scheele et al., 2013 Netherlands **Design:** Prospective longitudinal study **Funding Source** Department of GPErasmus MC, Rotterdam, Netherlands, Coolsingel Foundation, Rotterdam, The Netherlands | To determine the course of back pain in older patients and identify prognostic factors for non-recovery at 3 months' follow-up. | **N** = 675 **Mean age (SD)** 66.4 (7.6) Years **Female** = 401 (60%) **Missing Data** = 12% **LBP characteristics** • Duration: New episode • Distribution: Pain from top of scapula to first sacral vertebra **Comorbidities:** Not Reported | **Impairment based measure** • Finger-floor distance (cm) [C] • Quadriceps strength difference (yes, No) [D] **Performance based measure** • Timed up and Go test (sec) [C] **Equipment** Tape measure | Baseline | **Non recovery (GPE scale)** Self perceived 7-point scale ranging from 'completely recovered' to 'worse than ever' Back pain Disability **Timepoints** 3 Months (short) | **Bivariate Analysis-Logistic Regression Analysis** **Non-Recovery** • Finger to floor distance is not significant as a prognostic factor at baseline for non recovery at 3 month follow up. OR 1.0 P value (0.42) 95% CI (1.0–1.0) • Lower Quadricep strength difference is significant as prognostic factor at baseline for lower non recovery at 3 months. Quadriceps strength difference (yes vs. no) OR 1.8 P value (0.03) 95%CI (1.1–3.1) • Lower Timed up and Go test is significant as prognostic factor at baseline for lower non recovery at 3 months. OR 1.1 P value (<0.01) 95%CI (1.1–1.2) **Multivariate Logistic Analysis** • Timed up and Go test baseline is significant as prognostic factor for non recovery at 3 months OR 1.1 P value (0.01) 95%CI (1.0–1.2) |
| Shen et al. 2018 Korea **Design** Prospective longitudinal study **Funding Source:** Ministry of Education, Science and Technology Korea | To examine the influence of HGS on surgical outcomes after surgery for patients with DLSS. | **N** = 172 **High HGS group** n = 124 **Mean age (SD)** 68.1 (9.2) Years **Female** 61 (49.2%) **Low HGS group** n = 48 **Mean age (SD)** 72.3 (6.6) **Female** 34 (70.8%) **LBP Characteristics** • Duration: Not Reported • Distribution: Degenerative lumbar spinal stenosis **Comorbidities:** Not Reported | **Impairment-Based measure** • Hand grip strength (HGS) [C] **Equipment** Dynamometer | Baseline | **Disability** ODI **Health related Quality of life (HRQOL)** EQ-5D **Pain** VAS for back VAS for leg **Timepoints** 3,6 months (Short & medium) | **Multivariate Analysis** **One-way analysis of covariance (ANCOVA)** **Disability (ODI)** • High HGS compared to the low HGS at baseline is not significant as prognostic factor for ODI at 3 months P 0.184 • High HGS compared to low GHS at baseline is significant as prognostic factor for good outcome [lower ODI] at 6 months P 0.012 **HRQOL (EQ-5D)** • High HGS compared to low HGS at baseline is not significant as prognostic factor for EQ-5D at 3 months P 0.069 • High HGS compared to low HGS at baseline is significant as prognostic factor for good outcome [higher EQ5D] at 6 months P 0.039 **Pain (VAS)** • High HGS compared to low HGS at baseline is not significant as prognostic factor for pain-VAS (back) and VAS (leg) at 3 months P 0.722 • High HGS compared to low HGS at baseline is not significant as prognostic factor for pain -back and leg pain at 6 months VAS (back) VAS (leg) P 0.681 Note: Age, BMI, Gender controlled |

*(Continued)*

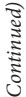

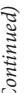

| Characteristics of study | Objective of study | Characteristics of participants | Potential physical prognostic factors | Baseline measurement timepoint | Outcome and measure, Outcome assessment timepoints | Results |
|---|---|---|---|---|---|---|
| Stolz et al. 2012 USA **Design** Prospective longitudinal study **Funding source** Not Reported | To derive a preliminary clinical prediction rule for identifying a subgroup of patients with low back pain (LBP) likely to benefit from Pilates-based exercise | **N** = 96 **Mean age (SD)** 56 (11.3) years **Female** = 81% **Missing Data** = 1 **LBP Characteristics** • Duration: <6 months 27%, >6mo 68% • Distribution: Lumbar spine 90.5% Buttock 68.4% Thigh 55.8% Lower leg/foot 32.6% **Comorbidities:** Not Reported | **Impairment based measures** • Total trunk flexion ROM, deg [C] • Pelvic flexion ROM • Lumbar flexion ROM • Total trunk extension ROM [C] • Right Side bending ROM deg Left side bending ROM deg • Average trunk side-bending ROM left and right, deg [C] • SLR left lower extremity, deg [C] SLR right lower extremity, deg • Right Hip rotation ROM deg [C] Left Hip rotation ROM, deg • Passive lumbar extension test [C] **Performance based measure** • Extensor endurance test [C] • Side support test-sec (Right, left) [C] • Active sit-up test (sec) [D] **Equipment** Inclinometer | Baseline | **Successful outcome Disability** (success-less than 50% ODQ score) ODQ score **Timepoints** 8 weeks (short term) | **Bivariate Analysis-Chi square test Disability** • Total trunk flexion ROM is significant as prognostic factor at baseline for success (lower ODQ score) at 8 weeks P value (0.04) (significance level P value <or equal to 0.10) • Pelvic flexion ROM is not significant as prognostic factor at baseline for success (lower ODQ score) at 8 weeks. P value (0.12) • Lumbar flexion ROM is not significant as prognostic factor at baseline for success (lower ODQ score) at 8 weeks. P value (0.12) • Total trunk extension ROM is not significant as prognostic factor at baseline for success (lower ODQ score) at 8 weeks. P value (0.12) • Left side trunk bending ROM is not significant as prognostic factor at baseline for success (lower ODQ score) at 8 weeks. P value (0.10) • Right side trunk bending ROM is not significant as prognostic factor at baseline for success (lower ODQ score) at 8 weeks. P value (0.12) • Average side bending ROM left and Right is not significant as prognostic factor at baseline for success (lower ODQ) at 8 weeks. P value (0.07) • SLR left lower extremity is not significant as prognostic factor at baseline for success (lower ODQ) at 8 weeks. P value (0.90) • SLR right lower extremity is not significant as prognostic factor at baseline for success (lower ODQ) at 8 weeks. P value (0.93) • Right hip rotation ROM is not significant as prognostic factor at baseline for success (lower ODQ) at 8 weeks. |

*(Continued)*

PLOS One | https://doi.org/10.1371/journal.pone.0335535 October 28, 2025

**Table 2.** (Continued)

| Characteristics of study | Objective of study | Characteristics of participants | Potential physical prognostic factors | Baseline measurement timepoint | Outcome and measure, Outcome assessment timepoints | Results |
|---|---|---|---|---|---|---|
| | | | | | | P value (0.16)<br>• Left hip rotation ROM is not significant as prognostic factor at baseline for success (lower ODQ) at 8 weeks.<br>P value (0.08)<br>• Positive Passive lumbar extension test is not significant as prognostic factor at baseline for success at 8 weeks.<br>P value (0.83)<br>• Extensor endurance test is not significant as prognostic factor at baseline for success at 8 weeks.<br>P value (0.35)<br>• Side support test (left) is not significant as prognostic factor at baseline for success at 8 weeks.<br>P value (0.20)<br>• Side support test (right) is significant as prognostic factor at baseline for success at 8 weeks.<br>P value (0.02)<br>• Positive active sit up test is significant as prognostic factor at baseline for success (lower ODQ score) at 8 weeks.<br>P value (0.05)<br>**Multivariate Analysis- Binary Logistic Regression Model**<br>• Total trunk flexion ROM<70° at baseline is significant as prognostic factor for good outcome; success [low ODQ] at 8 weeks.<br>LR +6.04 95%CI (1.45–25.13)<br>• Side Support test right side at baseline is significant as prognostic factor for good outcome; success [low ODQ] at 8 weeks.<br>LR 2.17 95%CI (1.54–3,06)<br>• Right or left hip average rotation > / 25° at baseline is significant as prognostic factor for good outcome; success [low ODQ] at 8 weeks.<br>LR 1.81 95%CI (1.18–1.77)<br>• Positive active sit up test at baseline is significant as prognostic factor for good outcome; success [low ODQ] at 8 weeks.<br>LR 2.44 95%CI (1.06–5.66)<br>Note: Studies concluded only two factors of these are robust factors based on accuracy calculation Total flexion ROM <70°, Right or left hip average rotation > / 25° |

*(Continued)*

Table 2. (Continued)

| Characteristics of study | Objective of study | Characteristics of participants | Potential physical prognostic factors | Baseline measurement timepoint | Outcome and measure, Outcome assessment timepoints | Results |
|---|---|---|---|---|---|---|
| Stroyer et al. 2008 Denmark **Design** Prospective longitudinal study **Funding source** Foundation funds were received in support of this work (National Research Centre for the Working Environment) | To study if low level of physical fitness was associated with increased low back pain (LBP) intensity at 30-month follow-up | **N = 327** **Mean age** 46 **Female =** 271 (83%) **Missing Data =** 113 **LBP characteristics** • Duration: 12 months • Distribution: low back **Comorbidities:** Not Reported | **Performance- based measures** • Back Extension Endurance (Modified Sorensen Test) [C] • Back Flexion Endurance [C] **Equipment** Not Reported | Baseline | **Pain intensity** NRS **Timepoints** 30 months (Long term) | **Multivariate Analysis- logistic regression analysis** **Pain** • Isometric back extension endurance is not significant as prognostic factor at baseline for good outcomes [low pain intensity- NRS] at 30 months Separate Logistic Regression-LR P=0.067 **Bivariate Analysis- Separate logistic regression** • Low back extension endurance is significant as a prognostic factor for good outcomes [low pain intensity-NRS] at 30 months LR P value (0.12) (level of sig <0.20) • Back flexion endurance is not significant as a prognostic factor for outcomes [high pain intensity-NRS] at 30 months LR P value (0.33) |
| Takala et al. 2000 Finland **Design** Prospective longitudinal study **Funding source** Finnish Work Environment Fund. | To study the predictive value of a set of tests measuring the physical performance of the back in a working population | **N = 123** **Female n =** 46 Age below 54-year-old **LBP Characteristics** • Duration: more than 30 days • Distribution: Not Reported **Comorbidities:** Not Reported | **Impairment- based measures** • Maximal side bending (mean of left & right) mm [C] • Extension/flexion strength ratio **Performance- based measure** • Static endurance test (sec) (Biering-Sørensen) [C] **Equipment** Goniometer An apparatus consisting of a horizontal handlebar connected with a nonelastic rope through a hole in the supporting platform to a servocontrolled electric engine below | Baseline | **Pain status** Recovered Pain less than 30 days, persistent pain-pain more than 30 days **Medical consultation** **Sick leave** **Timepoints** 1,2 Years | **Bivariate Analysis-(ANCOVA)** **Pain status – Pain persistence pain more than 30 days)/ recovery (pain less than 30 days)** • Maximal side bending at baseline is significant as prognostic factor for pain status – recovery [pain less than 30 days] at 1 year P (0.05) Level of significance NS > 0.10 • Extension/flexion strength ratio with 30 deg/sec at baseline is not significant as prognostic factor for pain status-recovery [pain less than 30 days] at 1 year P (0.08) • Static endurance test (Biering-Sørensen) at baseline is not significantly associated with pain status-pain recovery [pain more than 30 days] at 1 years P (0.08) **Medical consultation** • Low extension/ flexion force with 30 deg/ at baseline ratio is significant as prognostic factor for earlier first medical consultation P (0.002) **Sick Leave** • Low extension–flexion force ratio at baseline is significant as the prognostic factor for sick leave at 1 year P (<0.05) |

(Continued)

| Characteristics of study | Objective of study | Characteristics of participants | Potential physical prognostic factors | Baseline measurement timepoint | Outcome and measure, Outcome assessment timepoints | Results |
|---|---|---|---|---|---|---|
| Takenaka et al. 2019 Japan **Design** Prospective longitudinal cohort **Funding** Nagono Medical Foundation | To clarify objective predictors of postoperative 6-minute walk distance (6MWD) in patients with LSS and to develop prediction equations. | **N = 113** **Mean (SD) age:** 69.7 (8.9) **Female n (%):** 34 (43.6) **Missing data:** 13 (11%) **LBP characteristics** • Duration: 1.6 ± 2.2 years. **Comorbidities:** Diabetes mellitus (nine patients, 11.5%), heart disease (six patients, 7.7%), and depression (one patient, 1.3%), LSS | **Impairment-based measures** • Trunk extension strength [C] • Trunk flexion strength [C] **Performance-based measures** • 6 min walk test (6MWT) [C] **Equipment** Dynamometer | Baseline | **6 min walk test** (6MWD) **Timepoints** 6 months after surgery (Medium term) | **Multivariate Analysis-Multiple Regression Analysis** **6 MWD** • Higher preoperative 6 MWD test at baseline is significant as prognostic factor for improved 6 MWD at 6 months B0.31, SE0.08, P value (<0.01) • Higher Trunk flexor strength(kg) at baseline is significant as prognostic factor for improved 6 MWD at 6 months B 0.44, SE0.10, P value (<0.01) • Higher Trunk extensor strength(kg) at baseline is significant as prognostic factor for improved 6 MWD at 6 months B0.40, SE0.11, P value (<0.01) **Adjusted estimates:** • Preoperative 6MWD(m), B0.31, SE0.08, P value (<0.01) • Trunk extensor strength(kg), B0.26, SE0.11, P value (<0.01) |
| Van den Hoogen 1997 Netherlands **Design** Prospective longitudinal study **Funding source** Grants from the Dutch Organization for Scientific Research | To identify prognostic indicators of the duration of low back pain in general practice and the occurrence of a relapse. | **N = 443** **Mean age (SD)** 43.9 (14.6) years **Female** (52%) **Missing Data** n = 175 (40%) **LBP Characteristics** • LBP -duration: Not Reported • LBP distribution: Pain in the back (or radiating from the back) in the area between Th12 and the gluteal fold. **Comorbidities:** Not Reported | **Impairment -based measures** • Lumbar flexion by modified Schober's test [C] | Baseline | **Time to recovery** **Timepoints** 12 months (long term) | **Bivariate Analysis** **Time to recovery** • Higher performance on Schober's test for flexion is significant as prognostic factor for early time to recovery at 12 months P value (< 0.05) |

*(Continued)*

**Table 2.** (Continued)

| Characteristics of study | Objective of study | Characteristics of participants | Potential physical prognostic factors | Baseline measurement timepoint | Outcome and measure, Outcome assessment timepoints | Results |
|---|---|---|---|---|---|---|
| Wittink et al. 2002 USA **Design** Prospective longitudinal study **Funding source** Saltonstall Fund for Pain Research (equipment) No organizational funding | To investigate the association of aerobic fitness (V'O2max) with pain intensity as reported by a sample of patients with chronic (LBP). | **N = 75** **Mean age (SD)** 39.9(8.1) Years **Female = 42 (56%)** **LBP characteristics** • Duration: more than 3 months • Distribution: Back Only 46% Radicular pain 35% Radiculopathy 19% **Comorbidities:** Not Reported | **Performance based measure** • Aerobic fitness, measured by Peak Vo2 [C] **Equipment** Motor-driven treadmill Indirect calorimetry | Baseline | **Pain intensity** (NRS) **Timepoints** After test (short) | **Multivariate Analysis- regression analysis** **Pain** • Aerobic fitness: peak Vo2 at baseline is not significant as prognostic factor for pain intensity (NRS) after treadmill test. P value (0.31) |

Notes: Controlled for age and gender.

**Table 3. Quality assessment using QUIPS tool.**

| Study [n = 42] | Study participation | Study attrition [Follow-up] | Prognostic factor | Outcome | Confound-ing factor | Analysis | Overall risk of bias[a] |
|---|---|---|---|---|---|---|---|
| Berg et al., 2020 | Low | Low | Low | Low | Low | Low | Low |
| Burton et al., 1991 | Low | Moderate | Low | Low | High | High | High |
| Campello et al., 2006 | Low | Low | Low | Low | Low | Moderate | Moderate |
| Christensen et al., 1999 | Low | Low | Moderate | Low | Low | Moderate | Moderate |
| Coste et al., 1994 | Low | Moderate | Moderate | Low | High | Low | High |
| Ekedahl et al., 2012 | Moderate | Low | Low | Low | Moderate | Low | Low |
| Enthoven et al., 2003 | Low | Low | Low | Low | Moderate | Low | Low |
| Felicio et al., 2017 | Low | Low | Low | Low | Moderate | Low | Low |
| Flynn et al., 2002 | Low | Low | Low | Low | High | Low | High |
| Ghent et al., 2020 | Low | Low | Low | Low | High | Low | High |
| Gilmore et al., 2019 | Low | Moderate | Low | Low | Moderate | Low | Moderate |
| Gross et al., 2005 | Low | High | Low | Low | Low | Low | High |
| Grotle et al., 2007 | Low | Moderate | Low | Low | High | Low | Moderate |
| Gurcay et al., 2009 | Low | Low | Low | Low | High | Low | High |
| Hazard et al., 1991 | Moderate | High | Moderate | Moderate | High | Moderate | High |
| Hendrick et al., 2013 | Low | Low | Low | Low | Low | Low | Low |
| Hicks, et al., 2005 | Low | Moderate | Low | Low | High | Low | High |
| Hildebrandt et al., 1997 | Low | Moderate | Low | Moderate | High | Low | High |
| Hirayama et al., 2019 | Low | Moderate | Low | Low | High | Low | High |
| Hisamatsu et al., 2022 | Low | Low | Low | Low | Low | Low | Low |
| Ishibashi et al., 2023 | Low | Low | Low | Low | Low | Low | Low |
| Jain et al., 2023 | Low | High | Low | Low | Low | Low | High |
| Jakobsson et al., 2019 | Low | Moderate | Low | Low | Low | Low | Low |
| Karp et al., 2015 | Low | Low | Low | Low | High | Low | High |
| Kool.et al., 2002 | Low | Low | Low | Low | Moderate | Low | Moderate |
| Lagersted-Olsen et al., 2016 | Moderate | Moderate | Low | Low | Moderate | Low | High |
| Lee et al., 2017 | Low | Moderate | Low | Low | Moderate | Low | Moderate |
| Lubetzky et al., 2020 | Moderate | Low | Low | Low | Low | Moderate | Moderate |
| Mellin et at., 1993 | Moderate | Moderate | Low | Low | Moderate | Low | High |
| Milhouse et al., 1989 | High | High | Moderate | Low | High | High | High |
| Moradi et al., 2009 | Low | Moderate | Low | Low | Moderate | Low | Moderate |
| Nordeman et al., 2014, 2017 | Low | Low | Low | Low | Low | Low | Low |
| Patterson et al., 2022 | Moderate | Low | Low | Low | Low | Low | Low |
| Rodríguez-Romero et al., 2022 | Low | Low | Low | Low | Moderate | Low | Low |
| Scheele et al., 2013 | Low | Low | Low | Low | Moderate | Low | Low |
| Shen et al., 2018 | Low | Moderate | Moderate | Low | Low | Low | Moderate |
| Stolz et al., 2012 | Moderate | Low | Low | Low | High | Low | High |
| Stroyer et al., 2008 | Moderate | Moderate | Low | Low | Low | Low | Moderate |
| Takala et al., 2000 | Moderate | High | Moderate | Low | Low | High | High |
| Takenaka et.al., 2019 | Low | Low | Low | Low | Low | Low | Low |
| Van den Hoogen., 1997 | Low | Moderate | Low | Low | Low | Low | High |
| Wittink et al., 2002 | Low | Low | Low | Low | Low | High | High |

[a]Low RoB: If all six domains are rated as low RoB or no more than one is rated as moderate RoB.

High RoB: If one or more domains are rated as high RoB or ≥3 domains are rated as moderate RoB.

Moderate RoB: All studies in between are classified as moderate RoB.

**Table 4. Synthesis across included studies.**

| Physical prognostic factors | Study, country, risk of bias | Results | Summary of study findings [based on reported analysis-bivariate or multivariate analysis] | GRADE[b] quality of evidence | Summary of findings across studies |
|---|---|---|---|---|---|
| Handgrip strength (HGS) | Nordeman et al. 2017 Sweden Low RoB | **Bivariate Analysis** Handgrip strength (HGS) at baseline is not associated with activity limitation (RMDQ) at 2 years follow up Spearman rank correlation r (−0.17) P value (0.07) | Higher handgrip strength at baseline is not significant as prognostic factor for improved disability at long term | ++ Low | Using GRADE there is low quality evidence of no predictive ability of higher handgrip strength and improved disability outcome at long term |
| | Felicio et al. 2017 Brazil Low RoB | **Bivariate Analysis** Handgrip strength (HGS) at baseline is not associated with disability (RMDQ) at 12 months follow up Pearson correlation r (−0.16) P value (0.053) | Higher handgrip strength at baseline is not significant as prognostic factor for improved disability at long term | | |
| Isometric back extension endurance (seconds) | Enthoven et al. 2003 Sweden Low RoB | **Bivariate Analysis** Isometric back extension endurance at baseline is not associated with pain intensity (VAS) at 12 months follow up Univariate Regression Analysis- Simple Linear Regression B (−0.00) P value (> 0.05) | Higher isometric back extension endurance at baseline is not significant as prognostic factor for improved pain intensity at long term | ++ Low | Using GRADE there is low quality evidence of no predictive ability of higher isometric back extension endurance and improved pain intensity at long term |
| | Stroyer et al. 2008 Denmark Moderate RoB | **Bivariate Analysis** Isometric back extension endurance is not associated with pain intensity (NRS) at 30 months follow up Separate logistic regression-LR P value (0.067) | Isometric back extension endurance at baseline is not significant as prognostic factor for pain at long term | | |
| Fingertip-to-floor distance (centimeter) | Enthoven et al., 2003 Sweden Low RoB | **Bivariate Analysis** Fingertip-to-floor distance at baseline is not associated with pain intensity (VAS) at 12 months follow up Simple Linear Regression B (−0.03) P value (>0.05) R2 (0.00) | Higher fingertip-to-floor distance at baseline is not significant as prognostic factor for improved pain intensity at long term | ++ Low | Using GRADE there is low quality evidence of no predictive ability of higher finger tip to floor distance and improved pain outcome at long term |
| | Berg et al. 2022 Netherland Low RoB | **Bivariate Analysis** Fingertip-to-floor distance at baseline is not associated with pain intensity (NRS) at 12 months follow up Logistic regression analysis OR (1.2) P value (0.41) 95% CI (0.8–1.8) | Higher fingertip-to-floor distance at baseline is not significant as prognostic factor for improved pain intensity at long term | | |
| Isometric back flexion endurance (seconds) | Stroyer et al. 2008 Denmark Moderate RoB | **Bivariate Analysis** Isometric back flexion endurance at baseline is not associated with pain intensity (NRS) at 30 months follow up Separate Logistic Regression LR P value (0.33) | Higher isometric back flexion endurance at baseline is not significant as prognostic factor for improved pain intensity at long term | + Very Low | Using GRADE there is very low quality evidence of inconsistent predictive ability of higher isometric back flexion endurance and improved pain intensity at long term |
| | Enthoven et al. 2003 Sweden Low RoB | **Bivariate Analysis** Higher isometric back flexion endurance at baseline is associated with improved pain intensity (VAS) at 12 months follow up Simple linear regression B (0.13) P value (< 0.05) R2 (0.09) | Higher isometric back flexion endurance at baseline is significant as prognostic factors for good outcome; improved pain at long term | | |

*(Continued)*

| Physical prognostic factors | Study, country, risk of bias | Results | Summary of study findings [based on reported analysis-bivariate or multivariate analysis] | GRADE[b] quality of evidence | Summary of findings across studies |
|---|---|---|---|---|---|
| SLR[c]-ROM[d] (degree) (Passive-Max tolerance) | Hicks et al. 2005 US High RoB | **Bivariate Analysis** Higher SLR-ROM (bilateral) at baseline is associated with improved disability (ODQ) at 8 weeks follow up Regression Analysis LR (3.3) 95% CI (0.90–12.4) P value (0.069) Level of sig (P 0.10) | Higher SLR-ROM at baseline is significant as prognostic factor for good outcome; improved disability at short term | + Very Low | Using GRADE there is very low quality evidence of inconsistent predictive ability of SLR-ROM and improved disability at short term |
| | Stolz et al. 2012 USA High RoB | **Bivariate Analysis** SLR-ROM (bilateral) at baseline is not associated with success (dichotomized- improved ODQ score upto 50%) at 8 weeks follow up Chi-Square SLR-ROM (L) P-value (0.90), SLR-ROM(R) P-value (0.93) Level of sig (P 0.10) | Higher SLR-ROM at baseline is not significant as prognostic factor for improved disability at short term | | |
| Lumbar extension ROM (degree) | Hirayama et al. 2019 Japan High RoB | **Multivariate Analysis** Higher lumbar extension ROM at baseline is associated with improved disability (ODI) at 2 weeks follow up OR 5.39 95%CI (1.65–17.66) P value (0.005) Level of sig (0.15) | Higher lumbar extension ROM at baseline is significant as prognostic factor for improved disability at short term | + Very Low | Using GRADE there is very low quality evidence of inconsistent predictive ability of lumbar extension ROM and improved disability at short term |
| | Burton et al. 1991 UK High RoB | **Multivariate Analysis** Lumbar Extension ROM at baseline is not associated with disability at 1 month follow up No statistical values reported, only narrative statement provided | Extension ROM at baseline is not significant as prognostic factor for disability at short term | | |

[b]GRADE: Grading of Recommendations Assessment, Development and Evaluation

[c]SLR: Straight leg raise

[d]ROM: Range of motion

endurance (maximum time holding a prone position with trunk horizontal off the edge of a bench) and improved pain intensity (VAS and NRS) at 12 and 30 months. Therefore, low-quality evidence supports that higher isometric back extension endurance does not predict improved pain intensity long term.

**Inconsistent findings.** *Isometric back flexion endurance test with pain at long-term:* Very low-quality evidence (1 moderate and 1 low RoB studies [53,61]) supports inconsistent statistically significant associations between higher isometric back flexion endurance (maximum time holding the supine position with head and shoulders lifted) and improved pain intensity (NRS and VAS) at 12 and 30 months. Therefore, very low-quality evidence supports that higher isometric back flexion endurance does not consistently predict improved pain long term.

## Measures of activity in natural environment

No measures of activity in natural environment were synthesized due to wide variability of outcomes and follow-up time points across included studies.

## Potential prognostic factors in single studies

Across single studies, 23 physical measures of physical functioning out of 41 showed statistically significant associations with outcomes, indicating potential of predictive ability. A list of these measures in single studies is provided in the S7 File.

**Table 5. Adapted grading of recommendations assessment, development and evaluation (GRADE).**

**For Disability (ODQ, ODI) as an Outcome**

| Prognostic factor | Number of participants | Number of studies | Number of cohorts | Estimated effect size [95% CI] | Phase (design) | Factors that may reduce the quality | | | | | Factors that may increase the quality | | Overall quality |
|---|---|---|---|---|---|---|---|---|---|---|---|---|---|
| | | | | | | Study limitations | Inconsistency | Indirectness | Imprecision | Publication bias | Moderate/large effect size | Dose effect | |
| SLR-ROM | 150 | 2 | 2 | Unclear | 1 | ✗ | ✗ | ✓ | ✗ | ✓ | ✗ | ✗ | + Very Low |
| Lumbar extension ROM | 212 | 2 | 2 | Unclear | 1 | ✗ | ✗ | ✓ | ✗ | ✓ | ✗ | ✗ | + Very Low |
| *For Pain Intensity (VAS, NRS) as an outcome* | | | | | | | | | | | | | |
| Isometric back flexion endurance | 382 | 2 | 2 | Unclear | 1 | ✗ | ✗ | ✓ | ✗ | ✓ | ✗ | ✗ | + Very Low |
| Fingertip-to-floor distance | 598 | 2 | 2 | Unclear | 1 | ✓ | ✓ | ✓ | ✗ | ✓ | ✗ | ✗ | ++ Low |
| Isometric Back Extension Endurance | 382 | 2 | 2 | Unclear | 1 | ✓ | ✓ | ✓ | ✗ | ✓ | ✗ | ✗ | ++ Low |
| Handgrip Strength | 265 | 2 | 2 | Unclear | 1 | ✓ | ✓ | ✓ | ✗ | ✓ | ✗ | ✗ | ++ Low |

Twenty-two measures showed that higher measurement values predicted good outcome. In contrast, only one measure, 'time spent in sedentary,' indicated that a less sedentary time predicted a good outcome.

### Reporting bias

Three studies [57–59] were aligned with their registered protocols and 4 studies [55,60–62] had institutional review board-approved but unregistered protocols. No information was found for the remaining studies, contributing to risk of reporting bias.

## Discussion

The objective of this systematic review was to synthesize the evidence for physical measures of physical functioning as prognostic factors for predicting outcomes in the LBP population. The narrative syntheses highlighted the low-quality evidence for no predictive ability of higher FTF and higher isometric back extension endurance for improved pain intensity long term, and higher handgrip strength for improved disability long term. Very low-quality evidence supported inconsistent predictive ability of higher lumbar extension ROM and higher SLR-ROM for improved disability short term, and higher isometric back flexion endurance for improved pain intensity long term. Single studies identified 23 potential prognostic factors showing promising predictive ability for LBP outcomes. Variability in prognostic factors, outcomes and follow up timepoints hindered a comprehensive narrative synthesis.

### Consistent findings of no predictive ability

Low-quality evidence for no predictive ability of isometric back extension endurance for long-term pain aligns with the findings of a previous systematic review of Hartvigsen et al. [21] who found no association of isometric back extension endurance with short-term and long-term outcomes of disability and leg pain. Similarly, current review findings regarding the FTF test showing no predictive ability for long term pain, are consistent with the previous review findings [21]. Both reviews included prospective longitudinal studies; though the previous review was limited to the studies up to June 2012, the current systematic review includes studies from inception to May 2024 and also applies the GRADE approach, providing updated evidence and a more rigorous quality assessment. In this review, GRADE indicated that the low quality of supporting evidence is primarily due to imprecision, such as unreported or wide confidence intervals in the study results.

Low-quality evidence suggests no predictive ability of HGS for LBP outcomes in this review. To the best of our knowledge, no prior systematic review has explored the predictive ability of HGS in the LBP population. However, HGS has been shown to have predictive value in other populations. An umbrella review [64] demonstrated the predictive ability of HGS for various health outcomes, in cardiovascular disease, chronic kidney conditions, diabetes, and in the general population. This contrast between the low-quality evidence of no predictive ability in LBP and highly suggestive (class 2 evidence) predictive value of HGS in other health conditions highlights the need for high-quality evidence in the LBP to better understand the predictive ability of HGS in this population.

While these consistent findings suggest no predictive ability across the different physical measures of physical functioning, it is important to note that the findings are supported by low-quality evidence, so caution is required to interpret these findings. To address this, future research should focus on generating high quality evidence by conducting an adequately powered prospective longitudinal study. This will help to understand the predictive ability of these measures in LBP.

### Inconsistent findings of predictive ability

Very low-quality evidence supported inconsistent predictive ability of passive SLR-ROM, lumbar extension ROM for short-term outcomes, and isometric back flexion endurance for long-term outcomes. In this review included studies used passive SLR-ROM to assess passive hip flexion ROM, but its predictive ability was unclear due to inconsistent findings. Previous systematic review findings [21] also found that hip flexion ROM did not show consistent association with short

and long term outcomes in LBP. In both systematic reviews included studies used different methods of measuring hip flexion ROM that may have given inconsistent results,

In this review very low-quality of evidence for spinal extension ROM demonstrated inconsistent predictive ability for disability in the short term which is similar to the findings of a review [63] that found low quality evidence for no consistent relationship between changes in lumbar extension ROM and changes in pain or activity limitation. Current review covers the broader LBP population, whereas the previous review [63] was restricted to only nonspecific LBP. Similarly, in this review, very low-quality of evidence for back flexion endurance demonstrated inconsistent predictive ability for long-term outcomes, mirroring the inconsistent associations reported for long-term outcomes in the previous review [21]. Despite including the same study design (prospective longitudinal studies) in both reviews, the very low-quality of supporting evidence in this review is primarily attributed to high RoB studies in the synthesized evidence. Inconsistency in the predictive ability may be due to the methodological differences across included studies, measurement techniques (for example lumbar ROM measured by flexicurve or inclinometer) imprecision, inadequately controlled confounding factors and inconsistent reporting of univariate and multivariable analysis. Considering these challenges, future research should focus on conducting a low RoB prospective longitudinal study accounting for potential moderating variables and assuring consistent standards for test application, outcomes measurement and more consistency in reporting of univariable and multivariable analyses that will lead to the required high quality of evidence.

### Single studies suggesting predictive ability of potential prognostic factors

This review highlights significant heterogeneity among single studies for outcomes, follow up timepoints and different measurement methods of prognostic factors, that hindered a comprehensive synthesis of findings. The diversity across single studies emphasizes a need for standardized approaches (standardized protocols and methodologies) for consistent comparisons in future research. Out of the 41 prognostic factors identified across the single studies, 23 emerging physical measures of physical functioning showed promising potential as predictors of outcomes. Notably, half of these studies investigating performance-based measures and measures of physical activity in natural environment had a moderate to low risk of bias, highlighting an increasing trend towards high quality research investigating these physical measures in LBP. Given this, further research on these promising performance-based measures and natural activity measures is needed, through a low RoB prospective longitudinal study to establish their predictive value in LBP.

### Strengths and limitations of review

The strengths of this review are its rigorous conduct and reporting in accordance with PRISMA, Cochrane and AMSTAR. Inclusion of prospective longitudinal studies, the gold standard for prognostic research, enabled optimal synthesis of existing evidence for predictive ability of physical measures of physical functioning [23]. While screening the articles no restrictions such as language or publication date were applied. Every step was performed in duplicate with two independent reviewers. There are however some limitations that reflect the weak body of evidence that currently exists. The majority of studies in the narrative synthesis were limited to bivariate analysis, despite multivariable analysis being a powerful tool as it captures complex interplay between variables [64]. Also, wide variability across included studies for follow up times and outcomes hindered a comprehensive narrative synthesis.

In this review, applying the modified GRADE tailored for prognostic factor research provided a structured and transparent way to assess and communicate the quality of evidence. However, we acknowledge that GRADE is fundamentally a quantitative tool, and its use in the context of narrative synthesis presents inherent limitations.

### Challenges and future directions in prognostic research

This systematic review highlights methodological challenges in prognostic research of physical measures of physical functioning in LBP. One of the challenges is variability in how outcomes are defined (e.g., change versus absolute scores,

recurrence versus trajectory), which adds to the heterogeneity and complicates synthesis and comparison across studies. Future research should adopt standardized outcome definitions to enhance consistency and comparability. Another challenge is that most of the studies reported only p-values without corresponding confidence intervals, restricting true interpretation of prognostic associations. Future work should ensure that effect estimates are accompanied by appropriate measures of precision. Additionally, studies lack clear reporting of symptom duration, hindering evaluation of its role in prognosis. Improved reporting of symptom duration is needed to enhance the quality of prognostic analyses. Reflecting further upon the results, we noted that the biologic rationale informing the selection of many of the identified prognostic factors has not been adequately theorized. Some could be inferred (e.g., lower range of motion reflecting more severe structural pathology) yet even here they have been largely assumed. Strong theoretical rationale, including biological plausibility, is a necessary component of causality and seems an opportunity to strengthen the overall work in this area, which is important for advancing the field.

## Conclusion

This rigorous systematic review highlights that existing literature regarding the predictive ability of physical measures of physical functioning in LBP lacks high-quality evidence. Low-quality evidence supports no predictive ability of higher isometric back extension endurance, higher handgrip strength, and higher fingertip-to-floor test for good LBP outcomes long term. Very low-quality evidence supports inconsistent predictive ability of higher lumbar extension ROM, higher SLR-ROM short-term, and higher isometric back flexion endurance long term for good LBP outcomes. Low/very low-quality evidence suggests caution while interpreting these results. Imprecision, high risk of bias, variability of measurement techniques across studies, lack of standardized protocols, and inadequately controlled confounding factors contributed to low/very low-quality evidence.

This review also identifies emerging potential prognostic factors (performance-based measures, measures of activity in natural environment) showing promising predictive ability, and highlighting an increasing trend of improved research quality in this area. An adequately powered, low risk of bias prospective longitudinal study using standardized measurement protocols and multivariable analysis is required to further investigate the promising predictive ability of physical measures of physical functioning in LBP. Future prognostic research should be grounded in strong theoretical rationale, including biological plausibility. In a body of literature that is highly heterogenous this systematic review is providing an initial robust synthesis, positioning the researchers to advance the field from strong basis of understanding.

## Supporting information

**S1 File. AMSTAR 2 score of previous systematic reviews.**
(DOCX)

**S2 File. Search strategy.**
(DOCX)

**S3 File. QUIPS domains and judgmental formula.**
(DOCX)

**S4 File. GRADE criteria.**
(DOCX)

**S5 File. List of non-english studies.**
(DOCX)

**S6 File. Reasons of excluded studies at full text screening stage.**
(DOCX)

**S7 File.** List of potential prognostic factors in single studies.
(DOCX)

**S1 Checklist.** PRISMA checklist.
(DOCX)

## Acknowledgments

Study authors thank Alanna Marson, Librarian Western university for helping in development of search strategies. **Patient and public involvement:** The spinal pain research Patient Partner Advisory Group (PPAG) in the School of Physical Therapy at Western University has shaped the research methodology focused on physical measures of physical functioning. Systematic review results have been discussed with the PPAG to inform future research initiatives.

## Author contributions

**Conceptualization:** Rameeza Rashed, David Walton, Katie Kowalski, Alison Rushton.

**Formal analysis:** Rameeza Rashed, Afieh Niazigharemakher.

**Methodology:** Rameeza Rashed, David Walton, Katie Kowalski, Alison Rushton.

**Supervision:** David Walton, Katie Kowalski, Alison Rushton.

**Validation:** David Walton, Katie Kowalski, Alison Rushton.

**Writing – original draft:** Rameeza Rashed.

**Writing – review & editing:** Rameeza Rashed, David Walton, Katie Kowalski, Alison Rushton.

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
