## [Decision Letter · Decision Letter 0]

25 Aug 2025

Thank you for submitting your manuscript to PLOS ONE. After careful consideration, we feel that it has merit but does not fully meet PLOS ONE’s publication criteria as it currently stands. Therefore, we invite you to submit a revised version of the manuscript that addresses the points raised during the review process.

Dear Rameeza,

Thank you for submitting your manuscript to PLOS One. I would like to apologise for the delay in getting back to you due to the difficulty of obtaining two reviewers in the past few months. I have reviewed the manuscript and I am in agreement with reviewer 1, who has captured my concerns as well. We are looking forward to receiving a revision to your manuscript.

Kind regards,

Adrian

We look forward to receiving your revised manuscript.

Kind regards,

Adrian Pranata, Ph.D

Academic Editor

PLOS ONE

Reviewers' comments:

Reviewer's Responses to Questions

**Comments to the Author**

1. Is the manuscript technically sound, and do the data support the conclusions?

Reviewer #1: Yes

2. Has the statistical analysis been performed appropriately and rigorously?

Reviewer #1: Yes

3. Have the authors made all data underlying the findings in their manuscript fully available?

Reviewer #1: Yes

4. Is the manuscript presented in an intelligible fashion and written in standard English?

Reviewer #1: Yes

Reviewer #1: Thank you for the opportunity to review this manuscript examining the predictive ability of physical measures of physical functioning in patients with low back pain. I think the focus on physical measures is appropriate and an area that has been somewhat ignored in the large body of prognostic research related to low back pain. The review is by definition very broad and while this comes with some advantages it also creates significant challenges in appropriately synthesising and summarising the findings. I feel the authors have done quite a good job of this but it also creates some important limitations in the manuscript. My specific comments and suggestions are listed below.

1. The setup for the manuscript, and the reporting and interpretation of results do not really distinguish between prediction and causal associations. I think this is really important and needs to be covered in the introduction. Is the focus purely on prediction or is it on identifying factors that may be causally related 2 outcomes. This is important as there is a suggestion that prognostic factors are definitely useful but this is not necessarily the case. Prognostic factors which are not on the causal pathway may not inform future treatment and there is actually very limited evidence in the back pain field of prognostic factors leading to better treatment/outcomes. Even the STarT back tool as the most widely used prognostic tool has largely demonstrated a lack of ability to improve outcomes. So I'm just suggesting the introduction is more balanced in this regards and clearer in the intent as I think the authors are interested in causal associations and in some ways that contributes to the focus on physical factors.

2. Somewhat related to the point above I don't necessarily agree that adjusting for confounders is important if the focus is purely prognostic. I know this is one of the elements in the quality rating tool used but it really doesn't align with a purely prognostic focus in my opinion.

3. I have some mild concerns with using GRADE as part of a narrative synthesis. I understand the need to give the readers some sense of the quality of the evidence but to be honest I worry that GRADE scores can be somewhat misleading in such a situation. I can accept the authors continuing with this approach but I would at least like to see it mentioned as a limitation and particularly exploring the aspects of GRADE that may or may not make much sense in this situation. Do the authors really feel the GRADE ratings are a good representation of the quality of the evidence?

4. In the introduction I found the setup of physical functioning as an outcome to be somewhat clunky and complicated. Please review this paragraph. Throughout the manuscript I continued to struggle with this concept of physical functioning in some ways as the outcomes appeared to include primarily pain and disability. In addition, I suggest not using the abbreviation PF, as this is not a standard abbreviation and almost looked more like predictive factor to me so I found it confusing to read.

5. At line 144 there appears to be a typographical error in the sentence describing statistical heterogeneity. Please modify this and also do not purely refer to I-squared as there is substantial literature suggesting this is an inadequate measure of statistical heterogeneity.

6. Please provide a short but more detailed explanation of the rules used to determine if narrative synthesis was appropriate. For example 7 studies reported on fingertip to floor but only two were included in the narrative synthesis. What are some of the key elements that led to this decision?

7. What was the justification for not doing a quantitative synthesis in the studies where a narrative synthesis was appropriate?

8. When the authors refer to inconsistent associations does this mean that one was positive and one was not, or that the associations were in the opposite direction? Please provide a little bit more clarity around this including in the results for the narrative synthesis. The interpretation would be quite different if the estimates went in the opposite direction compared to a situation where they went in the same direction but in one study it did not quite reach statistical significance. The way the results are reported relies very much on P values which the authors would know of the substantial limitations.

9. In the key tables is it possible to indicate if the predictors have been dichotomized or are analysed on a continuous scale.

10. At line 243 please indicate the denominator. In other words 23 of how many?

11. The authors talk about the challenges of heterogeneity and as per my introductory comment this is a real challenge in such a broad review. One of my concerns is that some of these physical measures may have quite different associations depending on the outcome. For example a certain physical measure may be positively associated with change in the outcome but negatively associated with final score. There are many examples of this in the literature. In addition, certain physical factors my predict future back pain recurrence but not the trajectory of the current back pain. I think this concept needs at least a brief mention.

12. Somewhat related to the point above how many of the physical factors that were looked at have a clear rational or biological basis for association with the outcome measure that was used. How would finger tip to flaw be expected to be associated with prognosis? Is it used as a marker of the current state of severity in which case a good score of fingertip to floor may indicate a less severe case of low back pain indicating a better final outcome, but equally indicating a lower change score due to a lower starting score. I know this gets very complex, but I do think some acknowledgement of this complexity is required, and it would be nice to recommend investigation of physical factors where there is a clear rationale and hypothesis for the direction of the association. Maybe this is one of the most important conclusions.

13. Was the duration of symptoms at the time of the physical factor being measured considered. I can see logical reasons by a certain physical measure may have quite a different association with outcome if it is measured in a person with acute short-lasting pain compared to if the same measure is used in a patient with long term persistent pain?

**Do you want your identity to be public for this peer review?** For information about this choice, including consent withdrawal, please see our Privacy Policy

Reviewer #1: No

---

## [Author Response · Author response to Decision Letter 1]

8 Oct 2025

Date: 21-09-2025

Subject: Response to academic editor and Reviewers

Dear Academic Editor and Reviewers

Thank you for your review and consideration of our manuscript titled “Physical measures of physical functioning as prognostic factors to predict outcomes in low back pain: a systematic review and narrative synthesis” (PONE-D-25-04784) for publication in PLOS ONE. We have carefully reviewed your comments and incorporated the required changes as below.

Review Comments to the Author

Comment # 1

The setup for the manuscript, and the reporting and interpretation of results do not really distinguish between prediction and causal associations. I think this is really important and needs to be covered in the introduction. Is the focus purely on prediction or is it on identifying factors that may be causally related 2 outcomes. This is important as there is a suggestion that prognostic factors are definitely useful but this is not necessarily the case. Prognostic factors which are not on the causal pathway may not inform future treatment and there is actually very limited evidence in the back pain field of prognostic factors leading to better treatment/outcomes. Even the STarT back tool as the most widely used prognostic tool has largely demonstrated a lack of ability to improve outcomes. So I'm just suggesting the introduction is more balanced in this regards and clearer in the intent as I think the authors are interested in causal associations and in some ways that contributes to the focus on physical factors.

Authors Response:

Distinction between predictive factors and causal factors has been added in the introduction in the manuscript [Page 4, Line 59] and incorporated below for quick reference.

“A prognostic factor is any indicator that can predict subsequent health outcome and provides insights into the likely progression of a condition [8,9]. However, prognostic factors may not directly cause the outcome; rather, they can be markers or indicators of risk without lying on the causal pathway [10]. Causal factors always have some predictive value, but prognostic factors do not necessarily represent underlying causes. This study focuses exclusively on identifying and synthesizing prognostic factors that can predict LBP outcomes, rather than establishing causation.”

Comment # 2

Somewhat related to the point above I don't necessarily agree that adjusting for confounders is important if the focus is purely prognostic. I know this is one of the elements in the quality rating tool used but it really doesn't align with a purely prognostic focus in my opinion.

Authors Response:

We agree that the primary aim of prognostic research is outcome prediction rather than causal inference, and that adjustment for confounders is not always essential in a purely prognostic context. However, as recommended by the Cochrane Prognosis Methods Group, the QUIPS tool for quality assessment includes consideration of study confounding as one of the domains to evaluate potential bias. This domain ensures that important confounders are reported and accounted for, which can reduce risk of bias and increase the reliability of the prognostic effect estimates. Accounting for confounders in the analysis may improve the accuracy of prognostic models by controlling for variables that might otherwise distort the association between prognostic factors and outcomes. Therefore, including confounder adjustment as part of quality assessment of a study aligns with established standards.

Comment # 3

I have some mild concerns with using GRADE as part of a narrative synthesis. I understand the need to give the readers some sense of the quality of the evidence but to be honest I worry that GRADE scores can be somewhat misleading in such a situation. I can accept the authors continuing with this approach but I would at least like to see it mentioned as a limitation and particularly exploring the aspects of GRADE that may or may not make much sense in this situation. Do the authors really feel the GRADE ratings are a good representation of the quality of the evidence?

Authors Response:

We have mentioned in the published protocol of this systematic review that we will use modified GRADE as advised for prognostic factor research, a modified GRADE proposed by Huguet et al. that is fit for the purpose for this study. The modified GRADE consists of six domains (phase of investigation, study limitations, inconsistency, indirectness, imprecision, publication bias) that determine quality of evidence. Cochrane recommends using GRADE to assess the certainty of evidence in systematic reviews, including those with narrative syntheses when meta-analysis is not possible. GRADE can provide a transparent and structured approach to rating evidence quality even in narrative contexts. Cochrane Handbook Chapter 12 and 14 mention that narrative syntheses should provide structured summaries of findings and can include GRADE assessments to help interpret confidence in the evidence. We have addressed these points in the manuscript [Page 10, Line 173]. At the same time, we acknowledge that GRADE is fundamentally a quantitative tool, and its use in the context of narrative synthesis presents inherent limitations. This information has been added in the manuscript [Page 32, Line 357] and incorporated below for quick reference.

“In this review, applying the modified GRADE tailored for prognostic factor research provided a structured and transparent way to assess and communicate the quality of evidence. However, we acknowledge that GRADE is fundamentally a quantitative tool, and its use in the context of narrative synthesis presents inherent limitations.”

Comment # 4

In the introduction I found the setup of physical functioning as an outcome to be somewhat clunky and complicated. Please review this paragraph. Throughout the manuscript I continued to struggle with this concept of physical functioning in some ways as the outcomes appeared to include primarily pain and disability. In addition, I suggest not using the abbreviation PF, as this is not a standard abbreviation and almost looked more like predictive factor to me so I found it confusing to read.

Authors Response:

PF has been changed to full form, “physical functioning” throughout the manuscript.

Our systematic review is focused on physical functioning measures as prognostic factors rather than outcomes. This paragraph has been revised enhancing the clarity around the concept of physical functioning in the introduction [Page 4, Line 65] and incorporated below for quick reference.

“Physical functioning is a fundamental aspect of health, defined by the Core Outcome Measures in Effectiveness Trials (COMET) Initiative as the impact of a disease or condition on physical activities of daily living, such as walking and self-care [11,12]. It is recognized as a multidimensional construct encompassing several interconnected domains, including bodily structures and functions, performance of physical activities, as well as social and role-related participation [13]. Limitations in one domain may impact others, contributing to a decline in quality of life (QOL) [14].

Physical functioning can be assessed through different forms, including standardized self-report like the physical functioning subscale of the Short-Form 36 (SF-36) [15,16], can be directly observed by a rater (e.g., 6-minute walk test) [17], or can be quantified in real-world settings through wearable devices like accelerometers [18]. Each offer different insights into physical function, such as the patient’s own self-perceptions in the case of self-rating scales, or activity in ecological settings in the case of accelerometers. IMMPACT recommends using both direct observation/quantification of activity in addition to participant self-report for a more fulsome evaluation of a participants’ physical function [18]. In this systematic review, we are focused on physical measures of physical functioning that can predict outcomes in LBP.”

Comment # 5

At line 144 there appears to be a typographical error in the sentence describing statistical heterogeneity. Please modify this and also do not purely refer to I-squared as there is substantial literature suggesting this is an inadequate measure of statistical heterogeneity.

Authors Response:

The suggested changes have been made in manuscript [Page 9, Line 153].

“Methodologically, almost half of the included studies were at high RoB, while others were at moderate or low RoB. Statistical heterogeneity was high as indicated by I² values >50% and reflected in wide variation in effect estimates across studies.”

Comment # 6

Please provide a short but more detailed explanation of the rules used to determine if narrative synthesis was appropriate. For example, 7 studies reported on fingertip to floor but only two were included in the narrative synthesis. What are some of the key elements that led to this decision?

Authors Response:

Seven studies reported fingertip-to-floor, only two were included in the narrative synthesis because they assessed the same outcome at comparable follow-up times. The other five differed in outcome definitions or follow-up duration. The details of rules that were used to determine the narrative synthesis have been added in the manuscript [Page 9, Line 151] and incorporated below for quick reference.

“Due to high clinical, methodological and statistical heterogeneity [43], data were not pooled quantitatively. Clinically there was variability in LBP population characteristics, coexisting conditions, outcomes and follow up timepoints. Methodologically, almost half of the included studies were at high RoB, while others were at moderate or low RoB. Statistical heterogeneity was high as indicated by I² values >50% and reflected in wide variation in effect estimates across studies. This precluded a meaningful quantitative synthesis (meta-analysis), so a narrative synthesis was conducted [40].”

Comment # 7

What was the justification for not doing a quantitative synthesis in the studies where a narrative synthesis was appropriate?

Authors Response:

As justified above (response to comment # 6), our review presented substantial clinical, methodological, and statistical heterogeneity, which precluded quantitative synthesis. The justification for not doing quantitative synthesis in the studies where a narrative synthesis was appropriate has been added in the manuscript [Page 9, Line 156] and incorporated below for quick reference.

“According to the Cochrane Handbook, conducting a meta-analysis in the presence of substantial heterogeneity can produce misleading or clinically meaningless results and reduce the interpretability of the findings.”

Comment # 8

When the authors refer to inconsistent associations does this mean that one was positive and one was not, or that the associations were in the opposite direction? Please provide a little bit more clarity around this including in the results for the narrative synthesis. The interpretation would be quite different if the estimates went in the opposite direction compared to a situation where they went in the same direction but in one study it did not quite reach statistical significance. The way the results are reported relies very much on P values which the authors would know of the substantial limitations.

Authors Response:

The response for this comment has been addressed in the manuscript [Page 9, Line 165] and incorporated below for quick reference.

“Findings were considered inconsistent if studies reported associations in different directions or if they differed in statistical significance (achieved vs. not achieved), particularly when confidence intervals were not reported and only p-values were provided.”

All details of included studies are provided in Table 2: Data extraction of 42 included studies.

This limitation was compounded by the fact that many included studies reported only p-values without confidence intervals, restricting interpretation. We have addressed this point in the challenges and future directions in the manuscript [Page 32, Line 366] and incorporated below for quick reference.

“Another challenge is that most of the studies reported only p-values without corresponding confidence intervals, restricting true interpretation of prognostic associations. Future work should ensure that effect estimates are accompanied by appropriate measures of precision.”

Comment # 9

In the key tables is it possible to indicate if the predictors have been dichotomized or are analysed on a continuous scale.

Authors Response:

This information has been added with each predictor in the Table 2, Data extraction of 42 included studies [Page 12].

Comment # 10

At line 243 please indicate the denominator. In other words, 23 of how many?

Authors Response:

This information has been added in the manuscript [Page 31, Line 339] and incorporated below for quick reference.

“Out of the 41 prognostic factors identified across the single studies, 23 emerging physical measures of physical functioning showed promising potential as predictors of outcomes”.

Comment # 11

The authors talk about the challenges of heterogeneity and as per my introductory comment this is a real challenge in such a broad review. One of my concerns is that some of these physical measures may have quite different associations depending on the outcome. For example, a certain physical measure may be positively associated with change in the outcome but negatively associated with final score. There are many examples of this in the literature. In addition, certain physical factors my predict future back pain recurrence but not the trajectory of the current back pain. I think this concept needs at least a brief mention.

Authors Response:

The response for this comment has been added in the manuscript [Page 32, Line 363] and incorporated below for quick reference.

“One of the challenges is variability in how outcomes are defined (e.g., change versus absolute scores, recurrence versus trajectory), which adds to the heterogeneity and complicates synthesis and comparison across studies. Future research should adopt standardized outcome definitions to enhance consistency and comparability.”

In a body of literature that is highly heterogenous this systematic review is providing an initial robust synthesis, positioning the researchers to advance the field from strong basis of understanding.

Comment # 12

Somewhat related to the point above how many of the physical factors that were looked at have a clear rational or biological basis for association with the outcome measure that was used. How would finger tip to flaw be expected to be associated with prognosis? Is it used as a marker of the current state of severity in which case a good score of fingertip to floor may indicate a less severe case of low back pain indicating a better final outcome, but equally indicating a lower change score due to a lower starting score. I know this gets very complex, but I do think some acknowledgement of this complexity is required, and it would be nice to recommend investigation of physical factors where there is a clear rationale and hypothesis for the direction of the association. Maybe this is one of the most important conclusions.

Authors Response:

Response for this comment has been added in the manuscript [Page 32, Line 371] and incorporated below for quick reference.

“Reflecting further upon the results, we note that the biologic rationale informing the selection of many of the identified prognostic factors has not been adequately theorized. Some could be inferred (e.g. lower range of motion reflecting more severe structural pathology) yet even here they have been largely assumed. Strong theoretical rationale, including biological plausibility, is a necessary component of causality and seems an opportunity to strengthen the overall work in this area which is important for advancing the field.”

Comment # 13

Was the duration of symptoms at the time of the physical factor being measured considered. I can see logical reasons by a certain physical measure may have quite a different association with outcome if it is measured in a person with acute short-lasting pain compared to if the same measure is used in a patient with long term persist

---

## [Editor Report · Decision Letter 1]

14 Oct 2025

Physical measures of physical functioning as prognostic factors to predict outcomes in low back pain: a systematic review and narrative synthesis

PONE-D-25-04784R1

Dear Dr. Rashed,

We’re pleased to inform you that your manuscript has been judged scientifically suitable for publication and will be formally accepted for publication once it meets all outstanding technical requirements.

Kind regards,

Adrian Pranata, Ph.D

Academic Editor

PLOS ONE

Additional Editor Comments (optional):

Thank you for submitting your detailed responses to the reviewers’ and editor’s comments, and for providing a thoroughly revised version of your manuscript. We appreciate the time and care taken to address each point comprehensively.

After reviewing your responses and the revised manuscript, I am pleased to note that you have satisfactorily addressed all substantive concerns raised by the reviewers:

1. The conceptual distinction between prognostic (predictive) and causal factors has been clearly articulated in the introduction, clarifying the scope and intent of the review.

2. The discussion around confounder adjustment appropriately aligns with the guidance of the Cochrane Prognosis Methods Group and is justified in the context of prognostic research.

3. The rationale for using the modified GRADE framework for prognostic factor evidence has been well explained, with appropriate acknowledgement of its limitations within a narrative synthesis.

4. The concept of physical functioning has been substantially clarified, with removal of the “PF” abbreviation improving readability.

5. Methodological transparency has been enhanced through clearer reporting of inclusion criteria for narrative synthesis, specification of continuous versus dichotomised predictors, and clarification of denominators and heterogeneity interpretation.

6. The discussion section has been strengthened to acknowledge heterogeneity, variability in outcome definitions, biological plausibility, and inconsistent reporting of symptom duration—important issues for advancing prognostic research in low back pain.

Overall, the revisions have markedly improved the manuscript’s conceptual clarity, methodological rigour, and interpretative depth. Only very minor editorial refinements may be required at the proof stage (for example, brief clarification on alternative indicators of heterogeneity beyond I²).

---

## [Editor Report · Acceptance letter]

PONE-D-25-04784R1

PLOS ONE

Dear Dr. Rashed,

I'm pleased to inform you that your manuscript has been deemed suitable for publication in PLOS ONE. Congratulations! Your manuscript is now being handed over to our production team.

Kind regards,

on behalf of

Dr. Adrian Pranata

Academic Editor

PLOS ONE